# CompBench: Benchmarking Complex Instruction-guided Image Editing

## Abstract

While real-world applications increasingly demand intricate scene manipulation, existing instruction-guided image editing benchmarks often oversimplify task complexity and lack comprehensive, fine-grained instructions. To bridge this gap, we introduce **CompBench**, a large-scale benchmark specifically designed for *complex instruction-guided image editing*. CompBench features challenging editing scenarios that incorporate fine-grained instruction following, spatial and contextual reasoning, thereby enabling comprehensive evaluation of image editing models' precise manipulation capabilities. To construct CompBench, We propose an MLLM-human collaborative framework with tailored task pipelines. Furthermore, we propose an instruction decoupling strategy that disentangles editing intents into four key dimensions: location, appearance, dynamics, and objects, ensuring closer alignment between instructions and complex editing requirements. Extensive evaluations reveal that CompBench exposes fundamental limitations of current image editing models and provides critical insights for the development of next-generation instruction-guided image editing systems.

## 1 Introduction

Recent advances in instruction-guided image editing have pursued user-friendly and efficient manipulation of visual content. While such systems aim to simplify complex editing workflows, real-world applications often demand intricate instructions including spatial relationships, appearance details, and implicit reasoning. This necessitates the development of models with comprehensive capabilities in visual grounding, contextual understanding, and complex reasoning, thereby presenting substantial challenges to existing methodologies. However, as demonstrated in Figure 2, existing instruction-guided image editing benchmarks, *e.g.*, Emu Edit (Sheynin et al., 2024), MagicBrush (Yang et al., 2022a), and ReasonEdit (Huang et al., 2024b), exhibit critical limitations in assessing these essential capabilities, primarily in three aspects:

**Lack of Scene Complexity.** A key limitation of current benchmarks is their insufficient scene complexity, which hampers the representation of intricate visual structures inherent in real-world images. This stems from two main factors.

First, the prevalent use of synthetic images from text-to-image generation models, such as Stable Diffusion (Rombach et al., 2022), in previous benchmark construction (Yu et al., 2024; Ma et al., 2024) results in scenes with sparse spatial layouts, limited foreground object diversity, minimal occlusions, and simplistic textures and lighting conditions. Such artificial compositions lack dense object interactions, natural clutter, and photorealistic qualities essential for evaluating practical editing capabilities. Even when incorporating real images from datasets, such as COCO (Lin et al., 2014), these benchmarks often present oversimplified scenarios with elementary compositions insufficient for evaluating models on complex spatial relationships and interactions among multiple objects.

This problem is further exacerbated by benchmark design choices, wherein creators often deliberately exclude highly complex scenes featuring heavy occlusions, intricate details, or dynamic elements due to the challenges they pose for ground truth construction. While this practice facilitates more controllable evaluation, it creates a concerning discrepancy between benchmark performance and real-world applicability.

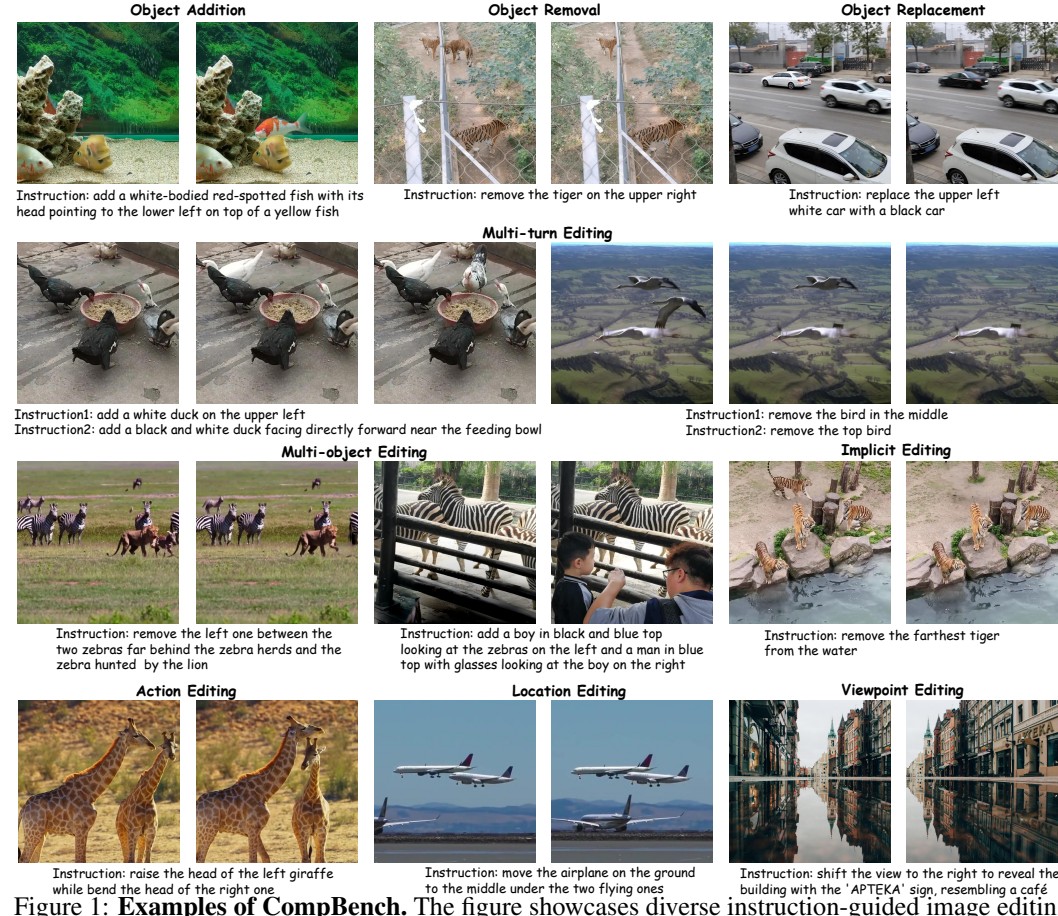

Figure 1: **Examples of CompBench.** The figure showcases diverse instruction-guided image editing tasks across nine categories: object addition, object removal, object replacement, multi-object editing, multi-turn editing, implicit reasoning, action editing, location editing and viewpoint editing.

Consequently, image editing models may attain high metric scores on these relatively simplified benchmarks, yet remain inadequate for real-world editing tasks that demand advanced scene understanding and manipulation. For instance, in reasoning-based tasks, InstructPix2pix (Brooks et al., 2023) exhibits a notable performance decline on our CompBench compared with ReasonEdit (Huang et al., 2024b), showing decreases of approximately 2.5 in PSNR, 0.02 in SSIM, and 0.4 in CLIP-Score.

**Limited Instruction and Task Comprehensiveness.** Beyond their oversimplified visual scenes, current benchmarks are further constrained by the narrow scope of editing instructions and tasks, failing to reflect the complexity of real-world user demands. Most existing datasets rely on simplistic, atomic-level instructions (*e.g.*, "change the dog to a cat") that lack contextual reasoning, and compositional logic typical of real user requests. In reality, user instructions often require complex reasoning and manipulation. These include multi-object editing ("remove the dog and the cat"), edits based on spatial relationships ("add a man to the right of the woman"), or action editing that modifies dynamic states ("make the man in white bend down more"). Current benchmarks, however, largely neglect these sophisticated task categories. This deficiency in instruction and task diversity prevents models from being rigorously tested on the full spectrum of challenges encountered in real-world applications. Consequently, their performance can be artificially inflated on simple tasks, providing an incomplete and misleading evaluation of true robustness and practical applicability.

**Deficiencies in Edited Image Quality.** Another critical limitation of current benchmarks is the suboptimal quality of their edited images. Many existing datasets exhibit two predominant issues that compromise their reliability: (1) instruction-alignment inaccuracies, where the edited output fails to precisely fulfill the specified modifications. (2) conspicuous visual artifacts, such as geometric distortions, background inconsistencies, or semantically incoherent objects. These quality deficiencies introduce substantial noise into performance evaluations, potentially leading to misleading assessments of model capabilities. Consequently, such benchmarks may fail to effectively discriminate between truly sophisticated editing systems and those that merely produce superficially plausible but flawed results.

Table 1: **Comparison of existing image-editing datasets and benchmarks.** Our benchmark supports seven core editing tasks, including multi-object, action and viewpoint editing, which are absent from most prior benchmarks. Scenario complexity is quantified by four indicators: *Avg. Obj.* (average number of objects per image), *Avg. Cat.* (average number of object categories per image), *OCC* (percentage of images that contain occluded objects), and *OOF* (percentage of images that contain out-of-frame objects). Details of these metrics can be found in Appendix C. Across all four metrics, our benchmark exhibits the highest complexity, underscoring its suitability for rigorous evaluation.

| Datasets / Benchmarks | Size | Types | Task | | | | | | | Complexity | | | |
|---|---|---|---|---|---|---|---|---|---|---|---|---|---|
| | | | Local | Multi-turn | Multi-obj. | Implicit | Action | Location | Viewpoint | Avg. Obj. | Avg. Cat. | Occ. Rate | OOF. Rate |
| *Datasets* | | | | | | | | | | | | | |
| InstructPix2pix (Brooks et al., 2023) | 313K | 4 | ✓ | ✗ | ✗ | ✗ | ✗ | ✗ | ✗ | 8.71 | 4.16 | 79.36 | 81.39 |
| EditWorld (Yang et al., 2024) | 8.6K | 1 | ✗ | ✗ | ✗ | ✓ | ✗ | ✗ | ✗ | 8.01 | 4.45 | 76.67 | 72.00 |
| UltraEdit (Zhao et al., 2024) | 4M | 9 | ✓ | ✗ | ✗ | ✓ | ✗ | ✗ | ✗ | 7.68 | 4.70 | 75.30 | 78.10 |
| SEED-Data-Edit (Ge et al., 2024) | 3.7M | 6 | ✓ | ✓ | ✓ | ✗ | ✗ | ✗ | ✗ | 6.21 | 3.82 | 63.82 | 81.40 |
| HQ-Edit (Hui et al., 2024) | 197K | 6 | ✓ | ✗ | ✗ | ✗ | ✗ | ✗ | ✗ | 8.22 | 4.84 | 66.97 | 60.30 |
| AnyEdit (Yu et al., 2024) | 2.5M | 25 | ✓ | ✗ | ✗ | ✓ | ✓ | ✓ | ✓ | 6.95 | 4.37 | 60.45 | 57.20 |
| ImgEdit (Ye et al., 2025) | 1.2M | 13 | ✓ | ✗ | ✗ | ✗ | ✓ | ✗ | ✗ | 9.01 | 4.72 | 69.65 | 69.14 |
| *Benchmarks* | | | | | | | | | | | | | |
| MagicBrush (Yang et al., 2022a) | 10K | 5 | ✓ | ✓ | ✗ | ✗ | ✓ | ✗ | ✗ | 9.22 | 5.04 | 91.71 | 78.34 |
| EMU_Edit (Sheynin et al., 2024) | – | 8 | ✓ | ✗ | ✗ | ✗ | ✗ | ✓ | ✗ | 8.38 | 5.19 | 78.51 | 83.60 |
| Reason-Edit (Huang et al., 2024b) | 0.2K | - | ✓ | ✗ | ✗ | ✓ | ✗ | ✗ | ✗ | 4.93 | 3.09 | 54.30 | 52.28 |
| I²EBench (Ma et al., 2024) | 2K | 16 | ✓ | ✗ | ✗ | ✓ | ✗ | ✗ | ✗ | 7.03 | 4.20 | 68.78 | 66.40 |
| GEdit-Bench (Liu et al., 2025) | 0.6K | 11 | ✓ | ✗ | ✗ | ✗ | ✓ | ✗ | ✗ | 9.96 | 4.93 | 67.67 | 65.40 |
| Complex-Edit (Yang et al., 2025) | 1K | 24 | ✓ | ✗ | ✗ | ✗ | ✓ | ✗ | ✓ | 9.23 | 4.77 | 78.29 | 72.98 |
| **Ours*** | 3K | 9 | ✓ | ✓ | ✓ | ✓ | ✓ | ✓ | ✓ | **13.58** | **5.87** | **98.47** | **86.38** |

Instruction: What would happen if the dog stumbled and it slid under snow?

Instruction: remove the motorcycle from the street

Instruction: change the color of fire hydrant to lavender

Instruction: replace the eagle with a parrot

Instruction: shift the horse in the image

Instruction: have a squirrel be looking at the vase

Instruction: change the red raspberry to a tangerine

Instruction: change the student to a professor

Instruction: make the pug a bulldog

Instruction: remove the white goose in the middle of the three geese

Instruction: add a red fish with black spots on the bottom left

Instruction: have the white cat jumping landing on the grass

Instruction: remove the second leftmost and the rightmost white goose

Instruction: make the two cars running on the road a little closer

Instruction: replace the man with a stripped shirt with a black person

Figure 2: **Comparison between current datasets or benchmarks and our CompBench. First row:** failed cases of other benchmarks. These results fail to maintain background consistencies or introduce noticable artifacts into the editing region. **Second row:** Examples of other benchmarks. These cases lack scene complexity and instruction comprehensiveness. **Third row:** Examples of our CompBench. Our benchmark features complex real-world scenarios with precise instructions.

To address the aforementioned issues, we introduce ***CompBench***, the first large-scale benchmark for instruction-based image editing in complex scenarios, specific examples are illustrated in Figure 1. Our benchmark offers the following three major advantages:

**Realistic and Complex Scene Composition.** As shown in Table 1, Our benchmark encompasses scenes that embody the diverse and realistic complexities present in real-world settings. We compare CompBench with existing datasets and benchmarks across four dimensions: average number of objects, average number of object categories, overall object occlusion rate, and out-of-frame object rate. Details of these metrics are shown in Appendix C. CompBench consistently surpasses prior benchmarks in all these metrics. Notably, our average number of objects per image is approximately **36.3%** higher than the second best (GEdit-Bench (Liu et al., 2025)), demonstrating the heightened complexity and diversity of our scenes.

**Comprehensive Task Coverage and High Difficulty Level.** As depicted in Figure 4(a), CompBench encompasses five major categories, consisting of local editing, multi-editing, action editing, scene

spatial editing, and complex reasoning, spanning a total of nine tasks. These tasks are designed to challenge six core capabilities, with a detailed analysis of our benchmark's difficulty for each provided in the Appendix B. Additionally, we propose an Instruction Decomposition Strategy to improve the clarity and precision of image editing instructions. Specifically, we structures editing instructions along four dimensions: spatial positioning (*e.g.*, "left of the table"), visual attributes (such as color or texture), motion states (*e.g.*, "flying"), and object entities. This structured approach converts potentially ambiguous requests into well-defined specifications without sacrificing the natural expressiveness of instructions. By systematically covering each aspect of an editing operation while preserving the flexibility of natural language, our method produces instructions that are both intuitively understandable and technically precise for complex image editing tasks.

**High-Quality Data Curation.** Every sample in CompBench is meticulously constructed through multiple rounds of expert review, ensuring the highest quality of edits. Unlike other benchmarks where editing failures are common, all data in CompBench represent successfully executed editing results, with SSIM (Structural Similarity Index Measure) scores significantly outperforming those of other datasets, as illustrated in Figure 4(b). This rigorous quality control ensures that CompBench provides a reliable assessment of model performance in realistically complex editing scenarios.

## 2 RELATED WORKS

**Instruction-guided Image Editing.** Instruction-guided image editing enables efficient image manipulation using only textual editing instructions, eliminating the need for manual mask or explicit visual inputs and better aligning with user intent. Diffusion models (Ho et al., 2020), particularly Stable Diffusion (Rombach et al., 2022) (SD), facilitate this task significantly by supporting explicit text inputs. Methods built upon diffusion models such as InstructPix2pix (Brooks et al., 2023), has greatly improved editing effectiveness. InstructPix2pix leverages large language models (LLMs) (Vaswani et al., 2017; Devlin et al., 2019; Brown et al., 2020; Touvron et al., 2023) and text-to-image (T2I) (Ramesh et al., 2021; 2022; Saharia et al., 2022; Rombach et al., 2022) models to generate large-scale datasets and trains a diffusion model that is capable of following natural language instructions. HIVE (Zhang et al., 2024) introduces a reward model that leverages human feedback to align edits with human preferences. Approaches such as SmartEdit (Huang et al., 2024b), MGIE (Fu et al., 2023), and Step1X-Edit (Liu et al., 2025) integrate image and instruction representations using multi-modal large language models (MLLMs) (Li et al., 2022; Alayrac et al., 2022; Liu et al., 2023; Wang et al., 2024), injecting these capabilities into diffusion models for more precise control. AnyEdit (Yu et al., 2024) constructs an extremely large-scale multi-task dataset and adopts a mixture-of-experts (MoE) (Fedus et al., 2022; Du et al., 2022) architecture to better accommodate diverse editing tasks. SEED-X (Ge et al., 2024) utilizes a visual tokenizer to unify image comprehension and generation, establishing a unified multi-granularity comprehension and generation model that enhances editing performance. GoT (Fang et al., 2025) incorporates Generation Chain-of-Thought (Wei et al., 2022) reasoning into the editing process, allowing for more refined, step-by-step edits. Recently, FLUX.1 Kontext (Labs et al., 2025) applies flow matching to build a unified image generation and editing model. Bagel (Deng et al., 2025) adopts a decoder only architecture to construct a multimodal understanding and generation model. Qwen-Image-Edit (Wu et al., 2025), the editing model of Qwen-Image (Wu et al., 2025), demonstrates strong text rendering and image editing capabilities.

**Image Editing Benchmarks.** High-quality image editing datasets and benchmarks are crucial for model training and evaluation. Several notable benchmarks have been proposed: MagicBrush (Yang et al., 2022a) provides a manually curated 10K dataset covering single-turn, multi-turn, mask-provided, and mask-free editing tasks. EMU-edit (Sheynin et al., 2024) introduces a challenging benchmark comprising seven diverse editing tasks. HQ-Edit (Hui et al., 2024) employs a scalable data collection pipeline to create a high-quality dataset of 200K instruction-guided image editing samples. SmartEdit (Huang et al., 2024b) introduces Reason-Edit, a small-scale, manually curated benchmark focused on complex instruction-based image editing. Edit-world (Yang et al., 2024) presents the concept of world-instructed image editing and creates a dataset featuring instructions in a world context. I2EBench (Ma et al., 2024) proposes a comprehensive evaluation benchmark with automated multi-dimensional assessment. UltraEdit (Zhao et al., 2024) develops a scalable framework for producing large and high-quality image editing datasets, introducing a large-scale instruction-based dataset. SEED-Data-Edit (Ge et al., 2024) provides a hybrid dataset composed of auto-generated, real-world, and human-annotated multi-turn editing samples. More recently, ImgEdit (Ye et al.,

Figure 3: **The construction pipeline of CompBench.** The pipeline consists of two main stages: (a) Source data collection and preprocessing, wherein high-quality data are identified through image quality filtering, mask decomposition, occlusion and continuity evaluation, followed by thorough human verification. (b) Task-specific data generation using four specialized pipelines within our MLLM-Human Collaborative Framework, where multimodal large language models generate initial editing instructions that are subsequently validated by humans to ensure high-fidelity, semantically aligned instruction-image pairs for complex editing tasks.

2025) introduces a large scale image editing dataset and a benchmark with multiple aspects. Step1X-Edit (Liu et al., 2025) construct GEdit-Bench (Liu et al., 2025) featuring real-world user instructions. Complex-Edit (Yang et al., 2025) adopts a "Chain-of-Edit" pipeline to develop an image editing benchmark across instructions of different complexity.

# 3 COMPBENCH

## 3.1 TASK CATEGORIZATION AND DEFINITIONS

Our complex instruction-guided image editing benchmark, **CompBench**, contains 3k+ image-instruction pairs. To enhance the comprehensiveness of evaluation, we categorize editing tasks into **five major classes** with **nine specific tasks** based on their characteristics:(1) Local Editing: focuses on manipulating local objects, including object removal, object addition and object replacement. (2) Multi-editing: addresses interactions among multiple objects or editing steps, including multi-turn editing and multi-object editing. (3) Action Editing: modifies the dynamic states or interactions of objects. (4) Scene Spatial Editing: alters scene spatial properties, consisting of location editing and viewpoint editing. (5) Complex Reasoning: requires implicit logical reasoning, including implicit reasoning. Examples of these tasks can be found in Figure 1.

## 3.2 DATASET GENERATION

In this section, we detailedly demonstrate the generation process of our CompBench. The overall pipeline is shown in Figure 3.

**Source Data Collection and Preprocessing.** To address the scarcity of high-quality paired complex editing data, we select MOSE (Ding et al., 2023), a video instance segmentation (VOS) dataset featuring complex scenes with multi-object masks. The dataset undergoes a rigorous preprocessing pipeline: We first filter low-quality video frames using a mixture of no-reference image quality assessment metrics (*e.g.*, NIQE (Zhang et al., 2015)) to eliminate blurry, low-contrast, or corrupted samples. Then, a professional team manually verifies the filtered data, retaining only high-quality images. For mask preprocessing, multi-object masks are decomposed into single-object masks to isolate editable entities. A multimodal large language model (MLLM, *e.g.*, Qwen-VL (Wang et al., 2024)) evaluates mask continuity and occlusion, discarding discontinuous or heavily occluded masks. Similarly, annotators further check these masks to ensure pixel-level precision.

**Task-specific Data Generation Pipelines.** To address the unique challenges and diversity of complex instruction-guided image editing tasks, we design four specialized data construction pipelines tailored to distinct task categories: (1) local editing pipeline for object-level manipulations (object removal, object addition, object replacement). (2) action/scene spatial editing pipeline for modifying object dynamics or scene perspectives (action editing, location editing, viewpoint editing). (3) complex reasoning pipeline for implicit contextual edits requiring reasoning (implicit reasoning). (4) multi-editing pipeline for multi-object and multi-turn editing tasks. All pipelines adopt a unified MLLM-Human Collaborative Framework: multimodal large language models (MLLMs) (Li et al., 2022; Alayrac et al., 2022; Liu et al., 2023; Wang et al., 2024) generate initial task-specific instructions by

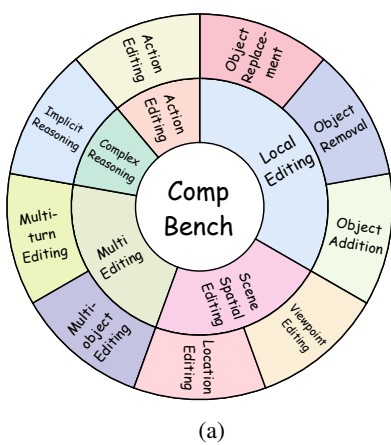 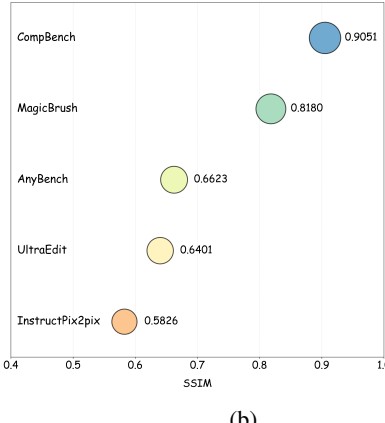

(a)                                         (b)

Figure 4: **Characteristics and statistics of CompBench.** (a) Task taxonomy of CompBench, illustrating the full range of task types. (b) SSIM (Wang et al., 2004) comparison among different datasets and benchmarks. Note that UltraEdit (Zhao et al., 2024) and InstructPix2pix (Brooks et al., 2023) are datasets, whereas the remaining entries are benchmarks.

analyzing visual scenes and editing goals, followed by human validation to ensure instruction-image semantic alignment and image editing fidelity. Unsuccessful edits are iteratively re-generated or discarded, retaining only high-fidelity samples that satisfy both linguistic precision and visual realism. Detailed implementation procedures for each pipeline are provided in the appendix A.

**Instruction Decomposition Strategy.** To enhance the clarity and precision of editing instructions, we propose a structured framework that organizes editing instructions along four aspects: spatial positioning, visual attributes, motion states, and object entities. This approach transforms ambiguous editing requests into well-defined specifications while maintaining natural expressiveness. The method employs a two-phase generation process: first, an MLLM produces dimension-aware instruction candidates by analyzing visual contexts. Then human experts refine these to ensure precision and consistency. By systematically addressing each aspect of the editing operation while preserving the flexibility of natural language, this framework enables the creation of instructions that are both intuitively understandable and technically precise for complex image editing tasks.

**Characteristics and Statistics.** As illustrated in Figure 4(a), our benchmark comprises 5 major categories encompassing a total of 9 complex editing tasks, yielding 3k+ image editing samples with corresponding complex instructions. Details of subtasks can be found in Appendix B.2. Since we lack global captions for pre-edited and post-edited images, we employ the Structural Similarity Index Measure (SSIM) (Wang et al., 2004) to evaluate the semantic consistency between image pairs as a quality assessment metric. As shown in Figure 4(b), CompBench achieves notably higher SSIM than other datasets and benchmarks.

Notably, our dataset features significantly more challenging editing tasks, each requiring comprehensive capabilities such as visual grounding and complex reasoning. Detailed analysis of core competencies essential for our benchmark are discussed in Appendix B. To systematically evaluate scene complexity, we adopt multiple quantitative indicators such as average number of scene objects and categories. These metrics demonstrate that our benchmark exhibits substantially higher complexity compared to existing benchmarks.

## 4 EXPERIMENTS

### 4.1 SETTINGS

**Baselines.** Given that our study specifically targets instruction-guided image editing tasks, we restrict our selection to instruction-guided image editing models and exclude approaches based on global description guidance. The evaluated models include: InstructPix2pix (Brooks et al., 2023), MagicBrush (Yang et al., 2022a), HIVE (Zhang et al., 2024), Smart-edit (Huang et al., 2024b), MGIE (Fu et al., 2023), HQ-Edit (Hui et al., 2024), CosXL-Edit (Stability AI, 2024), UltraEdit (Zhao

Table 2: **Evaluation results on local editing, multi-object editing and implicit reasoning.** LC-T denotes local CLIP scores between the edited foreground and the local description. LC-I refers to the CLIP image similarity between the foreground edited result and ground truth (GT) image. Top-three evaluation results are highlighted in **red** (1st), **blue**(2nd), and **green** (3rd).

| Model | Local Editing | | | | | Multi-object Editing | | | | | Implicit Reasoning | | | | |
|---|---|---|---|---|---|---|---|---|---|---|---|---|---|---|---|
| | Foreground | | Background | | | Foreground | | Background | | | Foreground | | Background | | |
| | LC-T ↑ | LC-I ↑ | PSNR(dB) ↑ | SSIM ↑ | LPIPS ↓ | LC-T ↑ | LC-I ↑ | PSNR(dB) ↑ | SSIM ↑ | LPIPS ↓ | LC-T ↑ | LC-I ↑ | PSNR(dB) ↑ | SSIM ↑ | LPIPS ↓ |
| InstructPix2pix (Brooks et al., 2023) | 19.445 | 0.777 | 21.416 | 0.695 | 0.137 | 19.624 | 0.779 | 20.200 | 0.658 | 0.162 | 19.007 | 0.793 | 21.806 | 0.683 | 0.125 |
| MagicBrush (Yang et al., 2022a) | 20.067 | 0.798 | 23.415 | 0.744 | 0.088 | 19.878 | 0.800 | 23.524 | 0.727 | 0.092 | 19.526 | 0.828 | 22.145 | 0.714 | 0.106 |
| HIVE-w (Zhang et al., 2024) | 19.820 | 0.770 | 19.908 | 0.641 | 0.198 | 20.024 | 0.775 | 19.594 | 0.609 | 0.224 | 18.634 | 0.777 | 20.268 | 0.602 | 0.219 |
| HIVE-c (Zhang et al., 2024) | 19.236 | 0.772 | 21.741 | 0.689 | 0.147 | 19.585 | 0.781 | 21.560 | 0.663 | 0.155 | 18.938 | 0.786 | 22.168 | 0.666 | 0.132 |
| Smart-edit-7B (Huang et al., 2024b) | 20.034 | 0.798 | 24.398 | 0.761 | 0.073 | 19.888 | 0.803 | 23.959 | 0.731 | 0.090 | 19.743 | 0.832 | 23.058 | 0.732 | 0.096 |
| MGIE (Fu et al., 2023) | 18.957 | 0.780 | 20.572 | 0.708 | 0.175 | 19.830 | 0.788 | 18.296 | 0.692 | 0.274 | 17.728 | 0.801 | 24.432 | 0.780 | 0.088 |
| CosXL-Edit (Stability AI, 2024) | 19.029 | 0.778 | 20.442 | 0.706 | 0.156 | 19.550 | 0.788 | 20.382 | 0.682 | 0.171 | 18.269 | 0.794 | 20.984 | 0.681 | 0.161 |
| HQ-Edit (Hui et al., 2024) | 18.316 | 0.734 | 12.240 | 0.419 | 0.441 | 19.163 | 0.757 | 12.987 | 0.412 | 0.421 | 18.864 | 0.767 | 12.321 | 0.396 | 0.452 |
| UltraEdit (Zhao et al., 2024) | 19.618 | 0.786 | 22.938 | 0.783 | 0.145 | 20.022 | 0.795 | 22.326 | 0.719 | 0.164 | 18.350 | 0.784 | 23.374 | 0.717 | 0.145 |
| AnyEdit (Yu et al., 2024) | 19.932 | 0.794 | 22.769 | 0.714 | 0.125 | 19.875 | 0.809 | 22.789 | 0.697 | 0.129 | 19.588 | 0.816 | 20.271 | 0.639 | 0.191 |
| SEED-X (Ge et al., 2024) | 17.933 | 0.780 | 21.466 | 0.805 | 0.139 | 19.092 | 0.795 | 20.638 | 0.788 | 0.158 | 17.467 | 0.784 | 21.506 | 0.709 | 0.134 |
| GoT (Fang et al., 2025) | 20.268 | 0.807 | 24.675 | 0.890 | 0.067 | 19.919 | 0.826 | 21.296 | 0.826 | 0.127 | 19.237 | 0.820 | 24.738 | 0.860 | 0.088 |
| Step1X-Edit (Liu et al., 2025) | 20.501 | 0.817 | 23.371 | 0.882 | 0.078 | 20.213 | 0.828 | 22.696 | 0.873 | 0.089 | 19.312 | 0.850 | 23.435 | 0.869 | 0.082 |
| Bagel (Deng et al., 2025) | 21.059 | 0.838 | 27.692 | 0.935 | 0.045 | 20.434 | 0.842 | 24.370 | 0.917 | 0.069 | 19.719 | 0.874 | 28.756 | 0.918 | 0.052 |
| FLUX.1 Kontext (Labs et al., 2025) | 21.329 | 0.821 | 25.612 | 0.941 | 0.049 | 20.983 | 0.836 | 24.013 | 0.938 | 0.064 | 19.606 | 0.867 | 25.330 | 0.932 | 0.061 |
| Qwen-Image-Edit (Wu et al., 2025) | 21.522 | 0.829 | 24.968 | 0.891 | 0.072 | 21.058 | 0.836 | 21.927 | 0.810 | 0.121 | 20.067 | 0.860 | 22.787 | 0.774 | 0.124 |

Table 3: **Evaluation results on multi-turn editing.**

| Model | Turn1 | | | | | Turn2 | | | | |
|---|---|---|---|---|---|---|---|---|---|---|
| | Foreground | | Background | | | Foreground | | Background | | |
| | LC-T | LC-I | PSNR | SSIM | LPIPS | LC-T | LC-I | PSNR | SSIM | LPIPS |
| InstructPix2pix (Brooks et al., 2023) | 19.424 | 0.784 | 21.073 | 0.676 | 0.142 | 19.818 | 0.776 | 17.607 | 0.568 | 0.238 |
| MagicBrush (Yang et al., 2022a) | 19.977 | 0.812 | 24.020 | 0.730 | 0.089 | 20.253 | 0.811 | 21.244 | 0.682 | 0.134 |
| HIVE-w (Zhang et al., 2024) | 19.784 | 0.781 | 20.040 | 0.621 | 0.196 | 20.129 | 0.761 | 17.291 | 0.532 | 0.272 |
| HIVE-c (Zhang et al., 2024) | 19.756 | 0.787 | 21.330 | 0.660 | 0.155 | 19.812 | 0.778 | 18.346 | 0.590 | 0.217 |
| Smart-edit-7B (Huang et al., 2024b) | 19.876 | 0.817 | 24.632 | 0.740 | 0.080 | 20.050 | 0.807 | 23.404 | 0.724 | 0.104 |
| MGIE (Fu et al., 2023) | 19.355 | 0.801 | 21.563 | 0.731 | 0.143 | 19.695 | 0.798 | 18.382 | 0.655 | 0.223 |
| HQ-Edit (Hui et al., 2024) | 18.987 | 0.755 | 12.950 | 0.410 | 0.422 | 18.935 | 0.740 | 12.032 | 0.383 | 0.499 |
| CosXL-Edit (Stability AI, 2024) | 19.389 | 0.787 | 20.233 | 0.679 | 0.171 | 19.394 | 0.771 | 16.752 | 0.590 | 0.301 |
| UltraEdit (Zhao et al., 2024) | 19.990 | 0.792 | 23.763 | 0.715 | 0.116 | 20.177 | 0.779 | 22.917 | 0.715 | 0.139 |
| AnyEdit (Yu et al., 2024) | 19.953 | 0.812 | 23.412 | 0.711 | 0.113 | 20.093 | 0.803 | 20.010 | 0.633 | 0.188 |
| SEED-X (Ge et al., 2024) | 19.139 | 0.795 | 21.042 | 0.792 | 0.153 | 18.729 | 0.753 | 13.793 | 0.457 | 0.404 |
| GoT (Fang et al., 2025) | 20.108 | 0.816 | 25.089 | 0.894 | 0.066 | 19.939 | 0.804 | 21.397 | 0.825 | 0.131 |
| Step1X-Edit (Liu et al., 2025) | 20.157 | 0.832 | 23.987 | 0.883 | 0.078 | 20.262 | 0.835 | 20.710 | 0.822 | 0.128 |
| Bagel (Deng et al., 2025) | 19.919 | 0.841 | 28.475 | 0.946 | 0.040 | 20.664 | 0.853 | 23.886 | 0.896 | 0.087 |
| FLUX.1 Kontext (Labs et al., 2025) | 20.061 | 0.837 | 25.725 | 0.951 | 0.050 | 21.172 | 0.843 | 22.357 | 0.906 | 0.095 |
| Qwen-Image-Edit (Wu et al., 2025) | 20.328 | 0.836 | 24.124 | 0.834 | 0.097 | 21.021 | 0.837 | 20.573 | 0.775 | 0.157 |

et al., 2024), AnyEdit (Yu et al., 2024), Seed-X-Edit (Ge et al., 2024), GoT (Fang et al., 2025), Step1X-Edit (Liu et al., 2025), Bagel (Deng et al., 2025), FLUX.1 Kontext (Labs et al., 2025), and the recently released Qwen-Image-Edit (Wu et al., 2025).

**Evaluation Metrics and Methods.** Evaluation metrics for image editing tasks must be well-suited to the complexity of our scenarios, providing a comprehensive and accurate assessment of editing performance in complex scenes. Moreover, the metrics should be tailored to reflect the unique characteristics of different task types.

For tasks including local editing, multi-editing and implicit reasoning, we posit that an effective editing model should modify foreground objects while preserving background consistency. Therefore, we apply a foreground-background decoupling strategy, evaluating the editing performance from both foreground and background perspectives. For background consistency assessment, we compute three metrics, including PSNR, SSIM (Wang et al., 2004), and LPIPS (Zhang et al., 2018), on the background regions. For the foreground evaluation, we consider two aspects: editing accuracy and instruction following. For editing accuracy, we measure the similarity between the edited result and the ground truth (GT) image by comparing their CLIP (Radford et al., 2021) image embeddings in the foreground region, thereby determining whether the edited foreground visually aligns with the GT. To assess instruction-following capability, we measure the CLIP (Radford et al., 2021) similarity between the edited foreground object and the textual description of the target region to evaluate the model's ability to interpret and execute the given instructions.

Additionally, for action editing, location editing, and viewpoint editing tasks—where the object's morphology, position, or viewpoint may change significantly—the aforementioned automatic metrics are insufficient for comprehensive evaluation. To address this, we introduce multi-perspective scoring using GPT-4o (OpenAI, 2024), Qwen2.5-VL-72B (Bai et al., 2025), and human annotators. For each task, we design tailored prompts for GPT-4o and Qwen-VL, instructing the models to rate editing performance on a scale from 0 to 10. In parallel, we conduct a rigorous human evaluation by trained annotators, following standardized scoring guidelines to measure aspects such as background fidelity, editing intent, instruction following, and artifact presence. Detailed prompt designs and annotation

Table 4: **Comparison on Action, Location, and Viewpoint Editing.** Results for GPT-4o, Qwen-72B, Human Evaluation, and Average scores (top-3 per column highlighted in red, blue, green).

| Model | Action | | | | Location | | | | Viewpoint | | | |
|---|---|---|---|---|---|---|---|---|---|---|---|---|
| | GPT | Qwen | Human | Avg. | GPT | Qwen | Human | Avg. | GPT | Qwen | Human | Avg. |
| InstructPix2pix (Brooks et al., 2023) | 3.047 | 1.124 | 3.101 | 2.424 | 3.425 | 2.167 | 2.859 | 2.859 | 0.699 | 0.482 | 0.036 | 0.406 |
| MagicBrush (Yang et al., 2022a) | 3.511 | 1.449 | 3.584 | 2.848 | 4.603 | 2.260 | 3.717 | 3.717 | 0.892 | 0.410 | 0.108 | 0.470 |
| HIVE-w (Zhang et al., 2024) | 3.151 | 1.764 | 3.067 | 2.661 | 4.110 | 2.192 | 3.421 | 3.421 | 1.494 | 0.283 | 0.036 | 0.604 |
| HIVE-c (Zhang et al., 2024) | 3.977 | 1.596 | 3.797 | 3.123 | 4.192 | 2.470 | 3.558 | 3.558 | 2.193 | 0.675 | 0.145 | 1.004 |
| Smart-edit-7B (Huang et al., 2024b) | 4.233 | 1.607 | 4.348 | 3.771 | 3.890 | 2.875 | 3.505 | 3.505 | 2.169 | 0.590 | 0.410 | 1.056 |
| MGIE (Fu et al., 2023) | 1.921 | 1.213 | 1.797 | 1.644 | 1.726 | 1.795 | 1.728 | 1.728 | 0.205 | 0.193 | 0 | 0.133 |
| CosXL-Edit (Stability AI, 2024) | 4.270 | 2.375 | 3.966 | 3.537 | 5.479 | 2.493 | 4.517 | 4.517 | 1.916 | 0.988 | 0.301 | 1.068 |
| HQ-Edit (Hui et al., 2024) | 1.449 | 0.528 | 1.033 | 1.003 | 1.425 | 0.726 | 1.079 | 1.079 | 0.470 | 0.289 | 0 | 0.253 |
| UltraEdit (Zhao et al., 2024) | 4.449 | 1.807 | 4.235 | 3.497 | 4.014 | 2.055 | 3.410 | 1.181 | 0.494 | 0.706 | 0 | 0.400 |
| AnyEdit (Yu et al., 2024) | 3.750 | 0.978 | 3.168 | 2.632 | 5.068 | 2.479 | 4.178 | 4.178 | 1.687 | 0.783 | 0.072 | 0.847 |
| SEED-X (Ge et al., 2024) | 2.270 | 1.494 | 1.685 | 1.816 | 3.028 | 3.247 | 2.771 | 2.771 | 2.241 | 1.169 | 0 | 1.137 |
| GoT (Fang et al., 2025) | 3.337 | 1.989 | 3.134 | 2.820 | 3.625 | 3.192 | 3.164 | 3.164 | 0.916 | 0.675 | 0.446 | 0.679 |
| Step1X-Edit (Liu et al., 2025) | 6.270 | 3.944 | 5.348 | 5.187 | 5.041 | 4.479 | 4.786 | 4.769 | 2.470 | 1.205 | 0.663 | 1.446 |
| Bagel (Deng et al., 2025) | 6.899 | 5.056 | 6.629 | 6.195 | 7.137 | 6.233 | 6.219 | 6.530 | 5.193 | 3.892 | 4.663 | 4.583 |
| FLUX.1 Kontext (Labs et al., 2025) | 5.169 | 3.202 | 4.517 | 4.296 | 3.000 | 3.110 | 3.836 | 3.996 | 3.471 | 2.373 | 3.108 | 2.984 |
| Qwen-Image-Edit (Wu et al., 2025) | 6.910 | 5.382 | 6.764 | 6.352 | 7.055 | 5.096 | 4.658 | 5.603 | 6.193 | 4.470 | 6.181 | 5.615 |

instructions are provided in Appendix G. Further ablation studies on evaluation metrics, as well as additional human evaluation results, can be found in Appendix D,H.

## 4.2 EXPERIMENT RESULTS

The experimental results for local editing, multi-turn editing, multi-object editing, implicit reasoning, and action/location/viewpoint editing are presented in Tables 2, 3, and 4, respectively. Our key analysis of the results are as follows: (1) No model dominates across all tasks. Among all evaluated models, Bagel (Deng et al., 2025) emerges as the most prominent one, achieving top results in 18 out of 37 metrics (nearly 60%) across 9 tasks. Notably, Bagel (Deng et al., 2025), Qwen-Image-Edit (Wu et al., 2025), and FLUX.1 Kontext (Labs et al., 2025) consistently deliver superior performance, securing top-three rankings in the majority of metrics across most tasks, following by Step1X-Edit (Liu et al., 2025). In contrast, HQ-Edit (Hui et al., 2024) demonstrates substantially inferior results in nearly all tasks. (2) For multi-turn editing tasks, all models exhibit a notable decline in background consistency metrics during the second editing round. Among them, SmartEdit (Huang et al., 2024b) maintains relatively robust performance in second editing turn. (3) Qwen-Image-Edit (Wu et al., 2025) achieves consistently high scores on the local CLIP scores between the edited foreground and the local description metric, reflecting its strong instruction-following and semantic alignment capabilities. In contrast, Bagel (Deng et al., 2025) ranks high on background consistency metrics, demonstrating its strength in preserving spatial and contextual background information during editing. (4) For the more challenging action/location/viewpoint editing tasks, Qwen-Image-Edit (Wu et al., 2025) and Bagel (Deng et al., 2025) perform comparably and significantly outperform most other models. Step1X-Edit (Liu et al., 2025) also exhibits promising editing performance in these scenarios.

## 5 INSIGHTS

In this section, we investigate the underlying factors that lead to varying performances among different models on our proposed CompBench, and offer perspectives on future research directions for the field of image editing.

**The Critical Role of MLLMs.** Through systematic evaluation, we discover a strong correlation between architectural design and editing performance: multi-modal large language models (MLLMs) (Li et al., 2022; Alayrac et al., 2022; Liu et al., 2023; Wang et al., 2024; Huang et al., 2024a; 2025; You et al., 2025; Li et al., 2025) serve as a cornerstone for recent advances in this field. Specifically, for a fair comparison, we normalize and average the performance scores of all models across five major tasks, highlighting the top-5 models in Figure 5(a). Furthermore, we present the overall normalized results in Figure 5(b), aggregated across all tasks. Details on the calculation of the results shown in the figure can be found in Appendix G. The results reveal that Bagel (Deng et al., 2025) significantly outperforms others on complex instruction-editing tasks, followed by Qwen-Image-Edit (Wu et al., 2025), FLUX.1 Kontext (Labs et al., 2025) and Step1X-Edit (Liu et al., 2025). Interestingly, the top-performing models, excluding the specialized flow-matching model FLUX.1 Kontext, either are MLLMs or integrate one as a core component. This architecture design empowers them to more accurately interpret complex instructions and visual context, which is critical for achieving superior performance on challenging editing tasks.

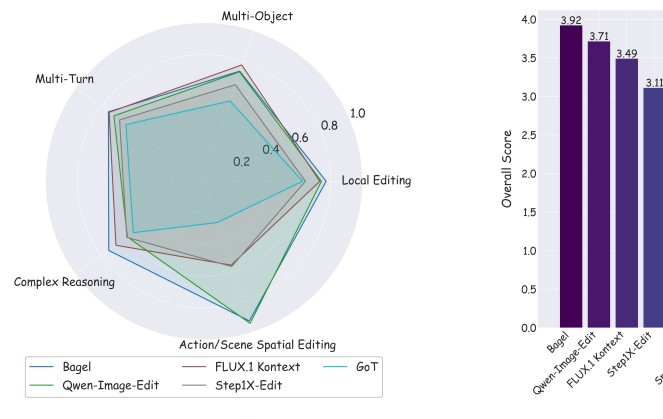

(a)                                                    (b)

Figure 5: **Overall Model Performance.** (a) Top 5 model performace in five major evaluation tasks. (b) Overall model performace across all tasks.

Moreover, we argue that MLLMs enable a unified understanding and generation paradigm. The best-performing model, Bagel, exemplifies this by jointly learning multimodal understanding and generation in a single model, yielding shared representations that are both instruction-aligned and visually grounded. This joint training reduces the mismatch between a planner (understanding model) and an executor (generation model), enabling the model to first encode complex images and instructions into semantically coherent multimodal features, which then guide the pixel-space editing process with high fidelity.

**The Importance of Reasoning Ability.** In addition to MLLM-driven achitecture design, our analysis reveals that reasoning ability emerges as another critical contributor to editing performance. This is evident through two distinct strategies. The first is data-centric: SmartEdit, for example, is trained on the reasoning segmentation dataset from LISA, which significantly enhances its reasoning capabilities and leads to outstanding results on multiple tasks. The second is method-centric: GoT introduces Chain-of-Thought (CoT) (Wei et al., 2022) into the editing process by leveraging MLLMs to generate reasoning chains. This approach further enhances the model's understanding of complex instructions and visual context, facilitating more precise editing.

In summary, our analysis reveals two critical insights for advancing instruction-guided image editing. Fisrt, MLLMs are pivotal for high-performance editing, providing a unified framework to bridge the gap between complex textual instructions and visual content. Second, multi-modal reasoning is foundational for interpreting intricate user intent to ensure high-fidelity edits. These findings suggeest that future research should prioritize two key directions: (1) developing specialized MLLM architectures tailored for editing workflows rather than general-purpose vision-language tasks (2) exploring advanced reasoning-aware training paradigms, such as optimizing reasoning chains via reinforcement learning (RL) or leveraging dedicated reasoning datasets, to enhance editing precision and adaptability.

# 6 CONCLUSION

In this work, we introduce CompBench, the first large-scale benchmark specifically designed for comprehensive evaluation of instruction-guided image editing. Our meticulously constructed benchmark encompasses five major categories with nine specialized tasks targeting complex image manipulation scenarios, comprising over 3,000 high-quality image editing pairs with corresponding natural language instructions. We conduct extensive experimental evaluation across 16 state-of-the-art instruction-guided image editing models on all benchmark tasks to systematically assess the capabilities and limitations of contemporary editing systems and validate the efficacy of our evaluation framework. The experimental findings from CompBench not only reveal significant performance gaps in current models but also yield valuable insights that elucidate promising research directions for advancing next-generation image editing systems with enhanced reasoning abilities and fine-grained control.

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

## A  IMPLEMENTATION DETAILS

### A.1  SOURCE DATA COLLECTION AND PREPROCESSING

The primary data collection and preprocessing pipeline has been described in section 3.2. For initial quality assessment of MOSE data, we employed four no-reference metrics (NIQE (Zhang et al., 2015), MANIQA (Yang et al., 2022b), MUSIQ (Ke et al., 2021), and CLIPIQA (Wang et al., 2023)). All images were systematically evaluated using these metrics, with the scores subsequently normalized and equally weighted to compute a composite value for filtering purposes. In terms of mask preprocessing, we decomposed multi-object masks into discrete single-object representations and utilized a multimodal large language model (specifically Qwen-VL (Wang et al., 2024)) to quantitatively assess mask continuity and occlusion levels, with the corresponding prompts illustrated in Figure 6. The comprehensive data preprocessing workflow underwent rigorous multi-round review and verification by a dedicated team of domain experts, thereby ensuring the exceptional quality and reliability of the final dataset.

**Action**

make the neck of the two giraffes close to each other

As a Dynamic Transformation Evaluator, your primary function is to assess the quality and realism of an object's movement or action within a scene, using two images: an original version and an edited version where the object has performed some action. Users will provide these images alongside the description of the intended action. Your task is to evaluate whether the action appears natural and physically plausible, maintains visual coherence with the scene, preserves appropriate motion blur and deformation consistent with the action, and ensures the edited object maintains proper interaction with its surroundings (including shadows, reflections, and contact points). Strictly provide your evaluation in a dict format, rating the quality of the dynamic transformation on a scale from 0 to 10, with 0 meaning poorly executed action and 10 meaning perfectly executed action. For example: {"score": 10, "reason": "Explanation here."} Please focus solely on providing your assessment in this dictionary format, avoiding any additional comments or extraneous details. IMPORTANT: When comparing the images, look for any evidence of the described action, even if subtle. Consider partial success in your scoring - even minor action changes that maintain scene consistency should receive appropriate partial credit. Only score 0 if the images are completely identical or if there's absolutely no attempt to implement the specified action. DO NOT SUPPOSE THE ACTION IS ACTUALLY IMPLEMENTED

**Location**

move the bird close to the wood

As an Object Movement Evaluator, your primary function is to assess the rationality and integration of an object's new position within a scene, using two images: an original version and an edited version where the object has been moved. Users will provide these images alongside the description of the movement. Your task is to evaluate whether the object's new position obeys physical laws, maintains consistency with lighting and perspective, aligns with the overall context of the scene, and ensures background consistency between the original and the edited image. Strictly provide your evaluation in a dict format, rating the suitability of the object's new position on a scale from 0 to 10, with 0 meaning poor integration and 10 meaning excellent integration. For example: {"score": 10, "reason": "Explanation here."} Please focus solely on providing your assessment in this dictionary format, avoiding any additional comments or extraneous details. IMPORTANT: First verify if the two images show evidence of object relocation. Even if the change is subtle, if background consistency is maintained well, provide a score that reflects the quality of integration. Only score 0 if the images are completely identical or if there's no attempt to move the specified object as instructed. Consider partial success in your scoring - minor changes with good background consistency should receive appropriate partial credit.

**Viewpoint**

shift the view downward

As a Viewpoint Transformation Evaluator, your primary function is to assess the quality and realism of a scene viewed from a different angle, using two images: an original version and an edited version where the camera viewpoint has changed. Users will provide these images alongside the description of the intended viewpoint change. Your task is to evaluate whether the new viewpoint maintains consistent spatial relationships between objects, correctly reveals or occludes elements based on the new angle, preserves proper perspective and foreshortening, maintains consistent lighting and shadows appropriate to the new viewpoint, and ensures texture and detail consistency across surfaces now viewed from different angles. Strictly provide your evaluation in a dict format, rating the quality of the viewpoint transformation on a scale from 0 to 10, with 0 meaning poorly executed viewpoint change and 10 meaning perfectly executed viewpoint change. For example: {"score": 10, "reason": "Explanation here."} Please focus solely on providing your assessment in this dictionary format, avoiding any additional comments or extraneous details. IMPORTANT: First verify if the two images (original and edited) actually show the same scene from different viewpoints. If they appear to be different scenes entirely or the viewpoint change is not evident, score 0 and explain that no proper viewpoint transformation was detected.

Figure 6: **Prompts of Editing Evaluation.**

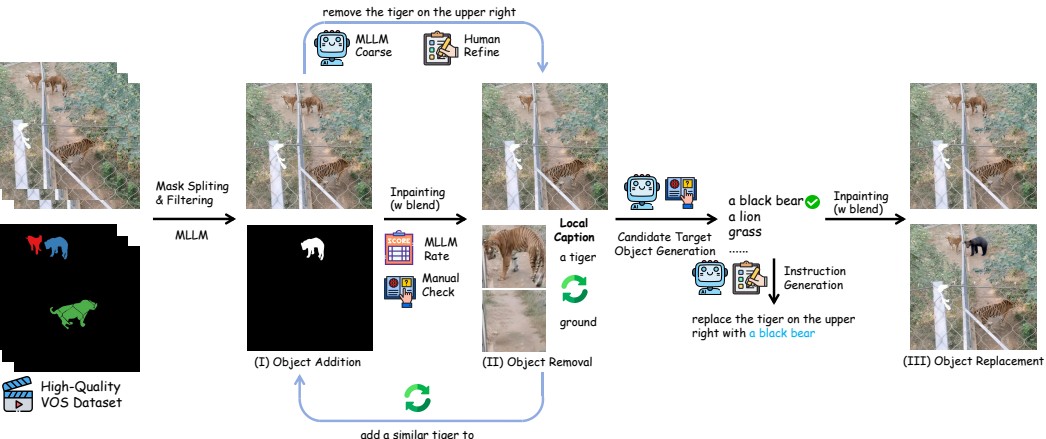

Figure 7: **Local Editing Pipeline.**

A.2   TASK-SPECIFIC DATA GENERATION PIPELINES

Due to the distinct characteristics of different tasks, we have designed specialized pipelines for data generation tailored to related task categories. The specific workflows for each pipeline are as follows:

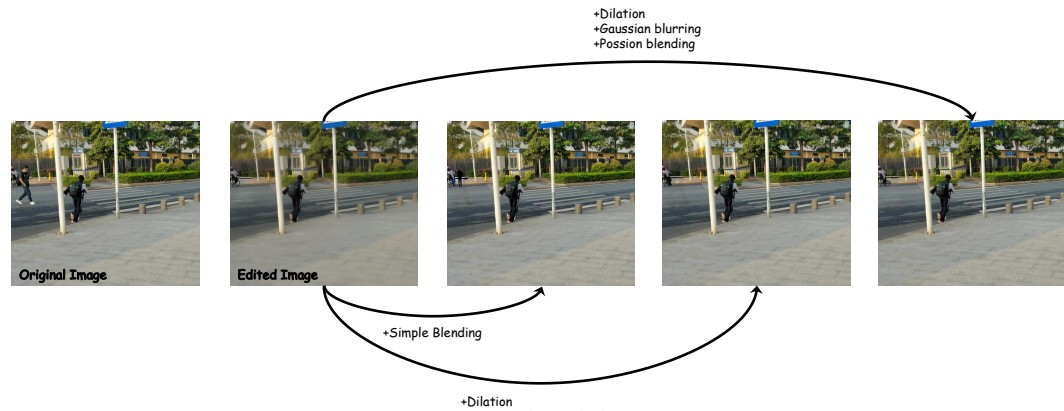

Figure 8: **Comparison of Different Blending Operations.**

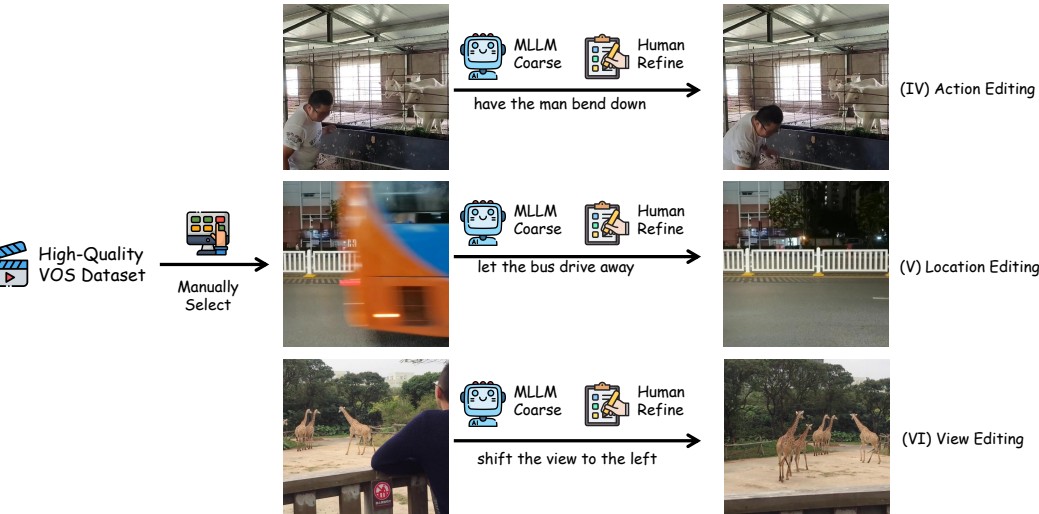

Figure 9: **Action/Scene Spatial Editing Pipeline.**

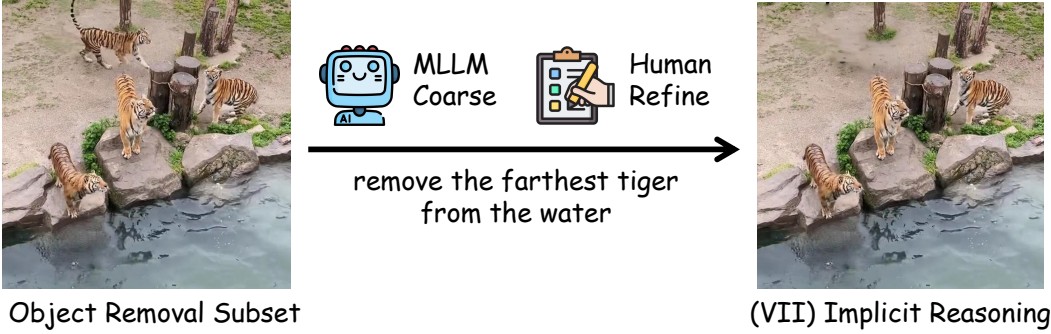

Figure 10: **Complex Reasoning Pipeline.**

**Local Editing Pipeline for Object-Level Manipulations.** As illustrated in Figure 7, given the preprocessed high-quality VOS dataset, we employ an inpainting model (PowerPaint (Zhuang et al., 2024)) to execute object removal based on precise object masks. The resultant outputs undergo rigorous evaluation and refinement through a Multimodal Large Language Model (MLLM) (Li et al., 2022; Alayrac et al., 2022; Liu et al., 2023; Wang et al., 2024) in conjunction with manual verification. For instruction generation, we provide the pre- and post-edited images alongside the corresponding masks, utilizing the MLLM to generate preliminary instructions, which are subsequently refined manually to ensure they accurately reflect the specific editing operations. To facilitate comprehensive

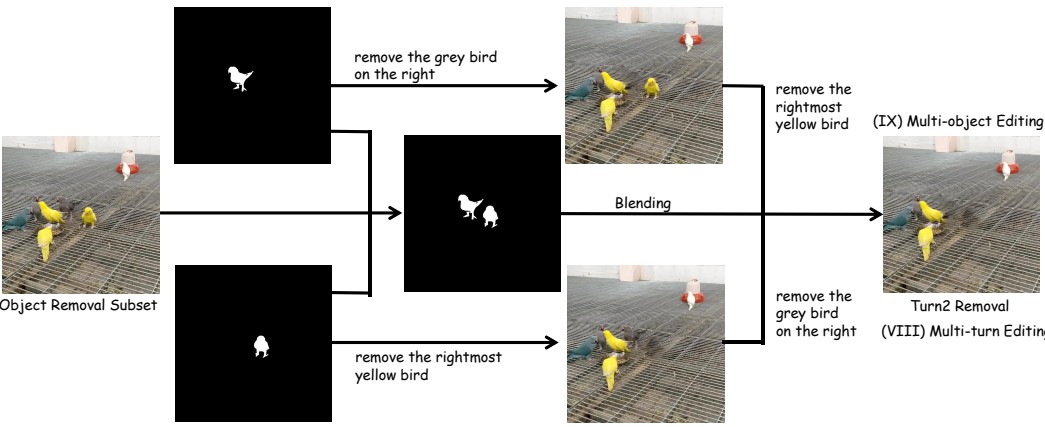

Figure 11: **Multi-editing Pipeline.**

evaluation, we systematically select and manually annotate captions for the mask regions before and after editing, thereby generating the definitive Object Removal dataset. For Object Addition data, we strategically reverse a subset of the Object Removal data and similarly employ MLLM complemented by manual annotations to construct the corresponding instructions. For Object Replacement, we implement a candidate target object generation approach, wherein the MLLM analyzes the before-and-after images and masks from Object Removal to propose plausible, diverse replacement objects that maintain contextual coherence with the scene's characteristics. Following manual selection of appropriate objects, we execute object replacement using the inpainting model, with instructions formulated analogously to those in the aforementioned tasks.

To further enhance background consistency and overall quality of inpainting results, we implement a composite post-processing strategy that integrates dilation, Gaussian blurring, and Poisson blending. Specifically, we first dilate the edges of the target object in the post-editing image, employing a kernel size of 20. Subsequently, Gaussian blurring is applied to the dilated edges with a kernel size of 15 and a $\sigma_X$ value of 3. Finally, Poisson blending is applied between the original image and the modified object region in the post-editing image to achieve the definitive result. As illustrated in Figure 8, the unprocessed edited images (direct inpainting outputs) exhibit noticeable blurriness and diminished clarity. Initially, we explored a rudimentary blending approach, directly merging the edited region with the background from the original image. However, this methodology resulted in pronounced and conspicuous boundaries, compromising the overall blending quality. By incorporating dilation and Gaussian blurring, the boundary integration improved substantially, yet unnatural chromatic discrepancies between the edited region and the surrounding background persisted. To address this limitation, we further integrated Poisson blending, which generated significantly superior results, yielding natural and seamless integration without perceptible artifacts.

**Action/Scene Spatial Editing Pipeline.** For action editing and scene spatial editing tasks (including location and viewpoint editing), as illustrated in Figure 9, we strategically select pertinent data from the VOS dataset. Specifically, for action editing, we extract frames from the same video sequence where the background remains consistent, while the objects undergo motion transformations. For location editing, we identify frames from identical video sequences where the background maintains constancy, but the object exhibits positional displacement. For viewpoint editing, we select frames exhibiting perspectival variations within the video. The generation of editing instructions for these tasks follows a methodology analogous to the local editing pipeline, integrating MLLM capabilities with meticulous manual refinement.

**Complex Reasoning Pipeline.** As depicted in Figure 10, for complex reasoning data, we systematically select a subset of data from the original VOS dataset that demonstrates suitability for implicit contextual edit instructions, subsequently modifying these instructions through manual intervention to produce the definitive dataset.

**Multi-Editing Pipeline.** For multi-object/multi-turn editing illustrated in Figure 11, we composite results from the object removal subset within the VOS dataset, where multiple object masks coexist within the same video frame, to generate comprehensive multi-object removal outputs. Conversely,

this approach establishes the foundation for multi-object addition. For multi-turn editing, the outcome from manipulating a single object constitutes an initial editing iteration, while the multi-object composite result represents a subsequent editing phase, culminating in the multi-turn removal outcome. Similarly, this procedural framework can be applied to multi-turn addition operations. All associated instructions undergo rigorous manual development and refinement to ensure precision and clarity.

## A.3 EXPERIMENT SETTING DETAILS

Table 5: **Configurations Details of Editing Models.**

| Method | Configuration |
|---|---|
| InstructPix2pix (Brooks et al., 2023) | guidance_scale=7.5
num_inference_steps=100
image_guidance_scale=1.5 |
| MagicBrush (Yang et al., 2022a) | guidance_scale=7.5
num_inference_steps=100
image_guidance_scale=1.5 |
| HIVE$^w$ (Zhang et al., 2024) | guidance_scale=7.5
num_inference_steps=100
image_guidance_scale=1.5 |
| HIVE$^c$ (Zhang et al., 2024) | guidance_scale=7.5
num_inference_steps=100
image_guidance_scale=1.5 |
| Smart-edit (Huang et al., 2024b) | guidance_scale=7.5
num_inference_steps=100
image_guidance_scale=1.5 |
| MGIE (Fu et al., 2023) | guidance_scale=7.5
num_inference_steps=100
image_guidance_scale=1.5 |
| HQ-Edit (Hui et al., 2024) | guidance_scale=7
num_inference_steps=30
image_guidance_scale=1.5 |
| CosXL-Edit (Stability AI, 2024) | guidance_scale=7.5
num_inference_steps=20
image_guidance_scale=1.5 |
| UltraEdit (Zhao et al., 2024) | guidance_scale=7.5
num_inference_steps=50
image_guidance_scale=1.5 |
| AnyEdit (Yu et al., 2024) | guidance_scale=3
num_inference_steps=100
image_guidance_scale=3 |
| SEED-X (Ge et al., 2024) | guidance_scale=7.5
num_inference_steps=100
image_guidance_scale=1.5 |
| GoT (Fang et al., 2025) | guidance_scale=5.0
num_inference_steps=50
image_guidance_scale=1.0 |
| Step1X-Edit (Liu et al., 2025) | guidance_scale=6
num_inference_steps=28 |
| Bagel (Deng et al., 2025) | guidance_scale=4.0
num_inference_steps=50
image_guidance_scale=2.0 |
| FLUX.1 Kontext (Labs et al., 2025) | guidance_scale=2.5
num_inference_steps=50 |
| Qwen-Image-Edit (Wu et al., 2025) | guidance_scale=4.0
num_inference_steps=50 |

**Evaluation Configurations.** We basically use the official settings of all models during evaluation. Specific configurations of each model are shown in Table 5.

**GPU Usage.** All experiments using the inpainting model to construct our benchmark were conducted on 8 NVIDIA A6000 GPUs (48 GB each). The same setup was used during evaluation to meet the memory and computation demands of all editing models.

## B   STATISTICS AND CAPABILITY ANALYSIS OF COMPBENCH

### B.1   CAPABILITY ANALYSIS

Table 6: **Task Competence Analysis.** We identify six core competencies essential for complex image editing. For each competency, the degree required by a task is classified as Low (L, 20), Medium (M, 50), High (H, 80), or Ultra High (UH, 100, bolded for extraordinarily high requirements). The last column reports the average competence score for each task, calculated by converting each required degree to its numeric score and then averaging.

| Task | Visual Grounding | Appearance Control | Relation Understanding | Complex Reasoning | Scene Consistency | 3D Geometry | Avg. Competence Score |
|---|---|---|---|---|---|---|---|
| Object Removal | H | M | H | H | **UH** | L | 68.3 |
| Object Addition | H | H | H | H | **UH** | L | 73.3 |
| Object Replacement | H | **UH** | H | H | **UH** | L | 76.7 |
| Multi-turn Editing | H | **UH** | H | H | **UH** | M | 81.7 |
| Multi-object Editing | **UH** | **UH** | **UH** | H | **UH** | H | 93.3 |
| Implicit Reasoning | H | H | **UH** | **UH** | **UH** | H | 90.0 |
| Action Editing | H | H | H | H | **UH** | H | 83.3 |
| Location Editing | **UH** | H | H | H | **UH** | M | 81.7 |
| Viewpoint Editing | H | H | H | **UH** | **UH** | **UH** | 90.0 |

To comprehensively characterize the requirements of complex image editing, we identify six core competencies that a sophisticated editing system must demonstrate: (1) **Visual Grounding.** the precise localization of target objects or regions; (2) **Appearance Control.** fine-grained manipulation of visual attributes such as color, texture, and illumination; (3) **Relation Understanding.** accurate modeling of semantic and spatial dependencies among objects; (4) **Complex Reasoning.** implicit logical deduction from contextual cues; (5) **Scene Consistency.** holistic preservation of spatial layout, occlusion patterns, and contextual coherence; and (6) **3D Geometry.** understanding and manipulating three-dimensional structure and viewpoint.

The quantitative correspondence between individual tasks and the competencies they necessitate is summarized in Table 6. Local editing tasks, such as Object Removal, Object Addition, and Object Replacement, place an exceptionally high premium on Scene Consistency, as seamless integration of the modified region is paramount. Multi-object Editing requires a balanced and very high proficiency in Appearance Control, Relation Understanding, and Scene Consistency to effectively manage complex inter-object interactions. Viewpoint Editing uniquely depends on the 3D Geometry competency to facilitate perspective transformations, as indicated by its ultra-high score in this domain. Meanwhile, Implicit Reasoning imposes stringent demands on Complex Reasoning and Relation Understanding to infer indirect or multi-step editing intentions.

Overall, CompBench presents substantial challenges by requiring the simultaneous integration of these multi-dimensional competencies, thereby reflecting the intricacy of real-world image-editing scenarios.

### B.2   STATISTICS OF COMPBENCH

Table 7: Number of edits for each editing task in the dataset.

| Object Removal | Object Addition | Object Replacement | Multi-turn Editing | Multi-object Editing | Implicit Reasoning | Action Editing | Location Editing | Viewpoint Editing | Total |
|---|---|---|---|---|---|---|---|---|---|
| 1331 | 982 | 152 | 576 | 144 | 100 | 89 | 73 | 83 | 3530 |

We present the detailed data volume for each task in Table 7 above. This allocation is informed by the competence analysis in Table 6. Specifically, we assign greater data volume to tasks with lower or moderate average competence scores, while allocating fewer examples to tasks with extremely high overall complexity. This design is motivated by three considerations.

First, all included tasks are significantly more challenging than those in existing image editing benchmarks. Since our focus is on complex scenarios, a dataset overloaded with high-difficulty tasks would likely suppress overall model performance, reducing score variance and impairing meaningful comparison. Second, for tasks with extremely high competence requirements, only a moderate number of samples is sufficient to robustly evaluate model capabilities—further increasing sample size yields diminishing returns in discrimination power. Third, collecting data for the most challenging tasks is substantially more resource-intensive, which naturally limits their quantity in the benchmark.

## C  DETAILS OF BENCHMARK COMPLEXITY EVALUATION

### C.1  COMPUTATION OF AVERAGE NUMBER OF OBJECT CATEGORIES, OBJECT INSTANCES, AND OCCLUSION RATE

To quantitatively evaluate the visual complexity and occlusion characteristics of images, we first defined a set of relevant metrics and criteria, and then employed an automated analysis pipeline powered by a Multi-modal Large Language Model (MLLM) (Alayrac et al., 2022; Li et al., 2022; 2023; Liu et al., 2023). Specifically, we utilized **Qwen2.5-VL-72B** (Wang et al., 2024), a state-of-the-art vision-language model capable of structured visual scene understanding. To ensure fairness, for datasets and benchmarks with more than 1,000 samples, we randomly sample 1,000 instances for evaluation.

The processing pipeline consists of the following steps:

1. Each image is encoded in Base64 format and input into the Qwen2.5-VL-72B model;
2. The model returns structured information in JSON format, which includes:
   - The total number of distinct object categories (`total_object_types`);
   - The total number of object instances in the image (`total_object_counts`);
   - The proportion of images containing occluded objects in the dataset (`occluded`).

After applying this process to the entire dataset, we remove statistical outliers to reduce bias. We then compute:

- **Average number of object categories per image**;
- **Average number of object instances per image**;
- **Average occlusion rate**, defined as:

$$\text{Occlusion Rate} = \frac{\text{Number of Images with at Least One Occluded Object}}{\text{Total Number of Images}}$$

### C.2  COMPUTATION OF THE OUT-OF-FRAME(OOF) METRIC

To evaluate whether objects are fully contained within image boundaries, we adopt a detection-based method using a pretrained object detection model. We employ the same random sampling strategy as described in Section C.1. The pipeline is as follows:

1. We apply **Grounding DINO** (Liu et al., 2024) to detect objects in each image and extract their bounding boxes, normalized by image dimensions;
2. Each bounding box is examined to determine whether it touches any of the four image boundaries (top, bottom, left, or right);
3. Objects whose bounding boxes contact any image edge are considered *not fully framed*;
4. For each image, we count the number of such boundary-touching objects and determine whether the image contains any incompletely framed object.

From this, we compute:

- **OFF Ratio**:

$$\frac{\text{Number of Images with at Least One Boundary-Touching Object}}{\text{Total Number of Images}}$$

# D  ABLATION STUDIES ON EVALUATION METRICS

## D.1  TARGETED EVALUATION ON COMPOUND NOUNS

Understanding compound entities remains a challenge for current vision-language models. Recent works (Kumar et al., 2024; Rambelli et al., 2024) point out that VLMs frequently struggle with interpreting compound nouns(e.g., "grassland"). To specifically assess the robustness of our automatic evaluation metrics in handling compound nouns (CNs), we performed targeted in-dataset experiments with both CLIP-based and GPT-4o-based evaluation.

**CLIP-Based Evaluation.** Approximately 15% of local captions in CLIP evaluation tasks contained compound nouns, representing a minor subset of the data. To analyze metric robustness, we randomly sampled 100 edited region–caption pairs featuring CNs, and replaced each compound noun with a plausible synonym (e.g., "grassland" → "meadow", "handrail" → "fence"). CLIP similarity scores were recalculated, and the distribution of absolute differences is shown in Table 8.

Table 8: CLIP Absolute Error Distribution After Synonym Substitution ($\leq 0.3$ considered negligible)

| Quantile (%) | 50 | 70 | 80 | 90 | 95 |
|---|---|---|---|---|---|
| Absolute Error | 0.19 | 0.26 | 0.30 | 0.48 | 0.80 |

The results show that up to the 80th percentile, score variations remain below the negligible threshold (0.3), although a few outliers at higher percentiles have larger differences.

**GPT-4o-Based Evaluation.** Compound nouns appeared in 6.9% of instructions requiring GPT-4o evaluation. For each, the compound noun was substituted by a plausible synonym, followed by re-evaluation of the edited results. All instances exhibited an absolute score difference $\leq 1$ point, indicating minor sensitivity to synonym changes.

**Discussion.** These results suggest that CLIP and GPT-4o metrics are generally robust to reasonable lexical variations in compound nouns, and can reliably assess compositional semantics in most cases. While CLIP may yield larger errors in a small number of complex cases, such outliers are rare and do not substantially affect metric reliability. GPT-4o was consistently robust in this evaluation. These findings support the validity of our automatic metrics for compound-noun scenarios.

## D.2  METRIC SENSITIVITY

To better understand the sensitivity of our automatic metrics, we conducted a targeted ablation study focusing on the Structural Similarity Index (SSIM). In particular, we investigated whether small changes in SSIM scores correspond to perceptually meaningful differences in background consistency.

Specifically, we randomly selected 50 cases in which the SSIM difference between editing results was within 0.02. For each case, three human annotators were tasked to compare the background consistency of each image pair and to judge which image presented better consistency. A match between the annotators' ranking and the SSIM score ordering was counted as a correct result. The average matching accuracy across all three annotators was 86%, suggesting that SSIM is generally sensitive to many visible background differences in our setting.

In addition, we re-evaluated the same image pairs with GPT-4o using tailored prompts regarding background consistency. Across these cases, GPT-4o's rankings matched SSIM-based rankings with an accuracy of 78%.

Further examination of disagreement cases revealed that purely perceptual evaluation can be ambiguous. For instance, discrepancies arose in situations when one background edit removed a visible object, while another retained all objects but had noticeable color differences. In such cases, human annotators sometimes disagreed, highlighting inherent limitations of both metric-based and manual assessments.

Overall, these findings indicate that SSIM drops—even as small as 0.02—often reflect perceivable image inconsistencies, while some edge cases may challenge both automated and human evaluation.

# E  MORE HUMAN EVALUATIONS

To further investigate the reliability and interpretability of our evaluation protocol, we conducted supplementary human experiments covering editing tasks evaluated with automatic metrics to jointly consider global semantic consistency, editing correctness, and perceptual quality across the entire image. Specifically, we collected human judgments for the following editing tasks: image addition, image removal, image replacement, multi-object editing, multi-turn editing, and implicit reasoning.

For each task, three representative models (a top-performing model, a moderate, and a weaker model) were selected for comparison. The mean human rating results for each task are summarized in Tabel 9 10 11 below:

Table 9: Local Editing: Human Evaluation Results

| Model | Addition | Removal | Replacement |
|---|---|---|---|
| InstructPix2pix | 0.794 | 0.784 | 2.530 |
| HQ-Edit | 0.021 | 0.034 | 0.039 |
| Step1X-Edit | 4.889 | 5.219 | 6.276 |

Table 10: Multi-Editing: Human Evaluation Results

| Model | Multi-Object | Multi-Turn (Turn1) | Multi-Turn (Turn2) |
|---|---|---|---|
| InstructPix2pix | 0.924 | 1.233 | 1.080 |
| HQ-Edit | 0.070 | 0.039 | 0.007 |
| Step1X-Edit | 4.799 | 5.705 | 5.403 |

Table 11: Implicit Reasoning: Human Evaluation Results

| Model | Reasoning |
|---|---|
| HQ-Edit | 0.320 |
| MGIE | 2.670 |
| Step1X-Edit | 5.560 |

To validate the reliability of our evaluation strategy, we compared these human ratings with the results from region-wise automatic metrics (see Table 2 and Table 3). The relative rankings and gaps among the three models are highly consistent between human and automatic assessments: for example, Step1X-Edit always leads in performance, HQ-Edit scores lowest, and InstructPix2pix or MGIE fall in between. This strong alignment indicates that our automatic region-wise metrics generally reflect human perception in distinguishing strong, moderate, and weak editing models.

# F  CASES OF COMPBENCH

In this section, we present additional exemplars from CompBench and comprehensive evaluation results. In Figure 12, we demonstrate representative instances of local editing operations (object addition, object removal, and object replacement) within our CompBench. In Figure 13, we illustrate selected cases of multi-turn and multi-object editing outcomes in our benchmark. In Figure 14, we showcase exemplary instances of action editing, location editing, and viewpoint editing capabilities within our benchmark.

We further present qualitative results from all evaluated models on our benchmarks. The comparative editing outputs across all models for local editing, multi-editing, and implicit reasoning tasks can be examined in Figures 15 and 16. The corresponding results for action editing, location editing, and viewpoint editing are displayed in Figures 17,18, and20. Detailed evaluation protocols and analytical discussions are presented in Section G.

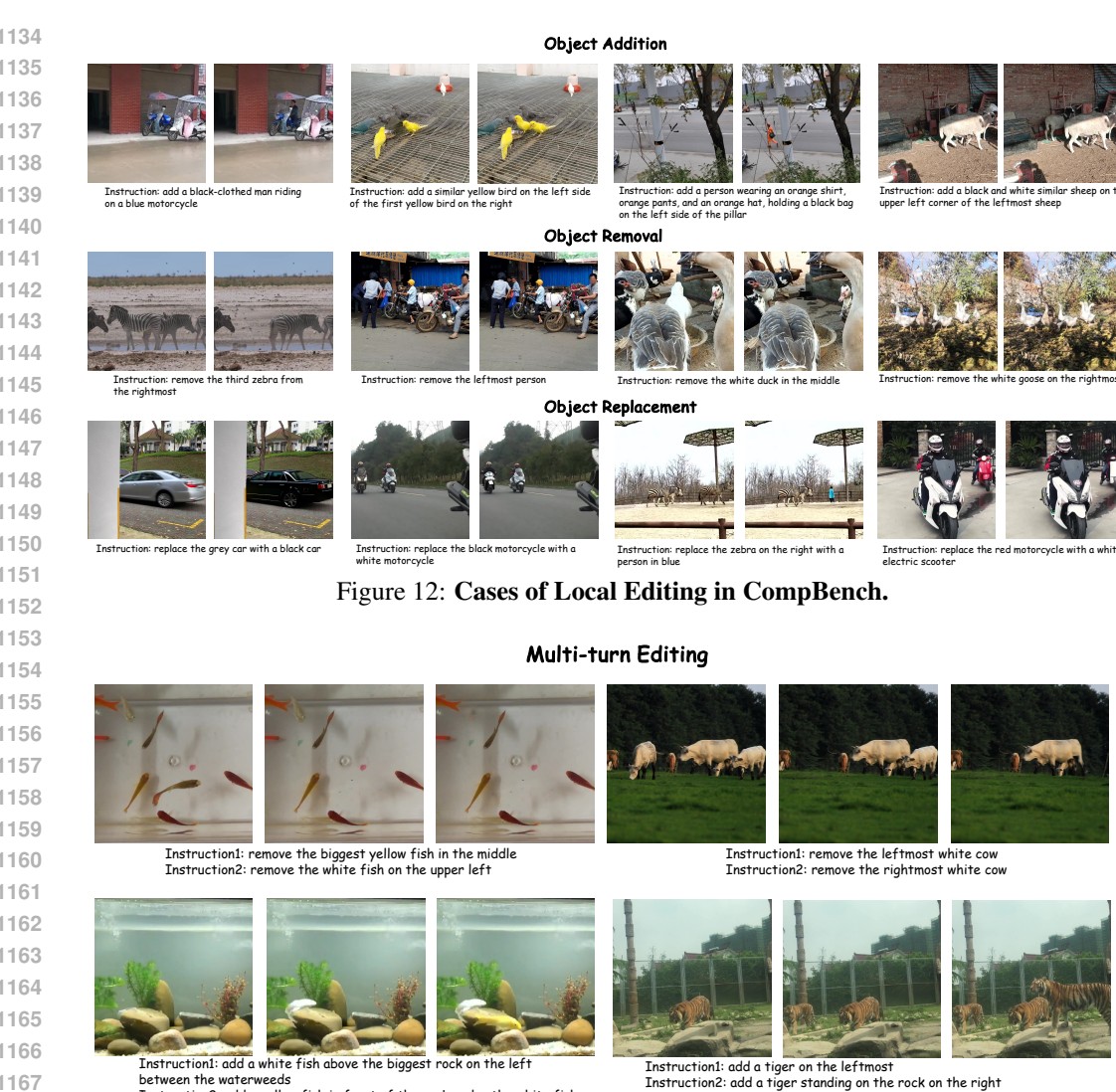

Figure 12: **Cases of Local Editing in CompBench.**

Figure 13: **Cases of Multi-editing in CompBench.**

## G EVALUATION DETAILS

In this section, we delineate the comprehensive evaluation methodology and procedural framework employed in our assessment.

For tasks encompassing local editing, multi-editing, and implicit reasoning, we require models to modify foreground elements while maintaining background fidelity. Consequently, we evaluate editing performance from both foreground and background perspectives. Background consistency is quantitatively assessed utilizing PSNR, SSIM, and LPIPS metrics on background regions, while foreground evaluation incorporates both editing accuracy via CLIP image embedding similarity with ground truth exemplars and instruction adherence via CLIP Score with the local foreground caption.

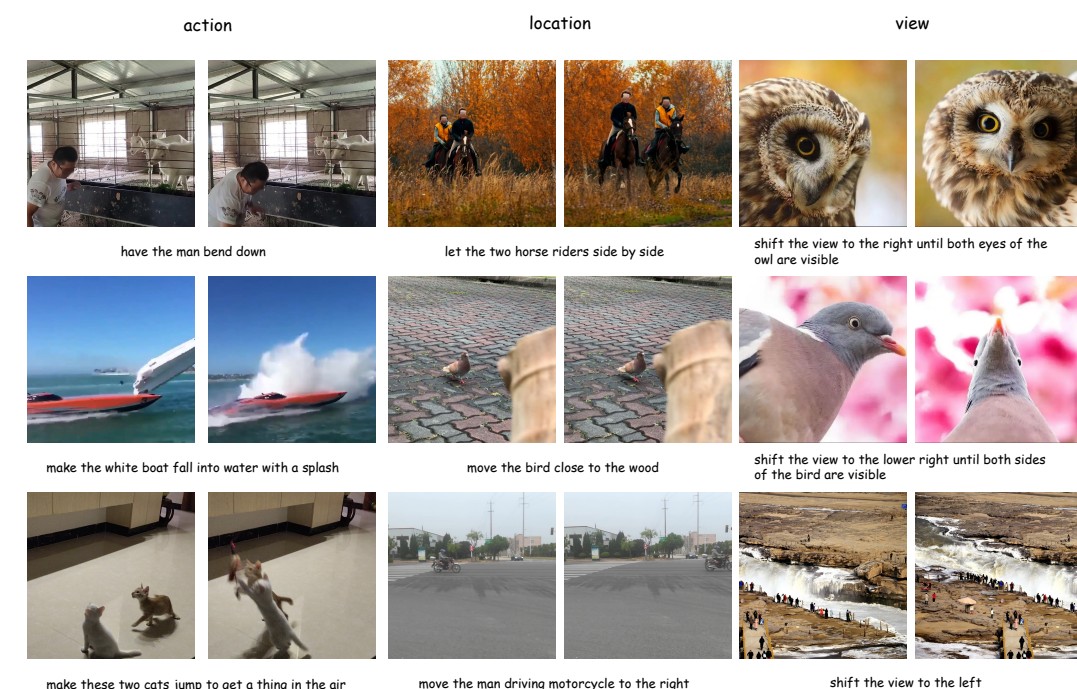

action                    location                    view

have the man bend down    let the two horse riders side by side    shift the view to the right until both eyes of the owl are visible

make the white boat fall into water with a splash    move the bird close to the wood    shift the view to the lower right until both sides of the bird are visible

make these two cats jump to get a thing in the air    move the man driving motorcycle to the right    shift the view to the left

Figure 14: **Cases of Action, Location and Viewpoint Editing in CompBench.**

The foreground captions are meticulously annotated and validated, with representative examples illustrated in Figure 12.

To objectively quantify the quality of Action Editing, Location Editing, and Viewpoint Editing, which require substantial image manipulations, we devise an automated evaluation pipeline leveraging GPT-4o and Qwen-VL: for each source–edited image pair, the pipeline generates a reproducible and interpretable composite score ranging from 0 to 10, accompanied by a concise textual justification. Initially, we formulate three task-specific prompts as demonstrated in Figure 6, each emphasizing two to three criteria extracted from the fundamental task characteristics: action-execution correctness, preservation of non-target regions, and overall realism for Action; positional accuracy, occlusion consistency, and global harmony for Location; and plausibility of viewpoint transformation, geometric coherence, and detail preservation for Viewpoint. All prompts instruct GPT-4o to return a standardized JSON object of the form {"score": <0-10>, "reason": "<...>"}. Subsequently, each image pair is processed by GPT-4o precisely once with the decoding temperature fixed at 0, yielding deterministic and consequently fully reproducible results.

To aggregate the performance across the diverse evaluation suite, a unified scoring methodology is employed. The evaluation encompasses five distinct tasks: local editing, multi-object editing, multi-turn editing, complex reasoning, and Action/Scene Spatial evaluation. For normalization, MinMax scaling is applied independently to the metrics within each task, converting them to a uniform 0-1 range. To align with a "higher is better" convention, LPIPS scores are inverted after normalization (i.e., 1-normalized value). The overall scores, presented in Figure 5(b), are then calculated by summing each model's normalized scores across all five equally weighted tasks. This results in a final composite score with a theoretical maximum of 5.0.

## H HUMAN ANNOTATIONS

**Annotation Details.** To complete our benchmark development, we engaged four expert annotators and one quality assurance specialist to supervise all human verification processes throughout the benchmark construction. This comprehensive oversight encompassed filtering the initial datasets, validating generated images, and most critically, formulating precise instructions. During the instruction annotation phase, annotators were provided with the original images, edited images, and corresponding masks, and were tasked with crafting instructions based on this multimodal infor-

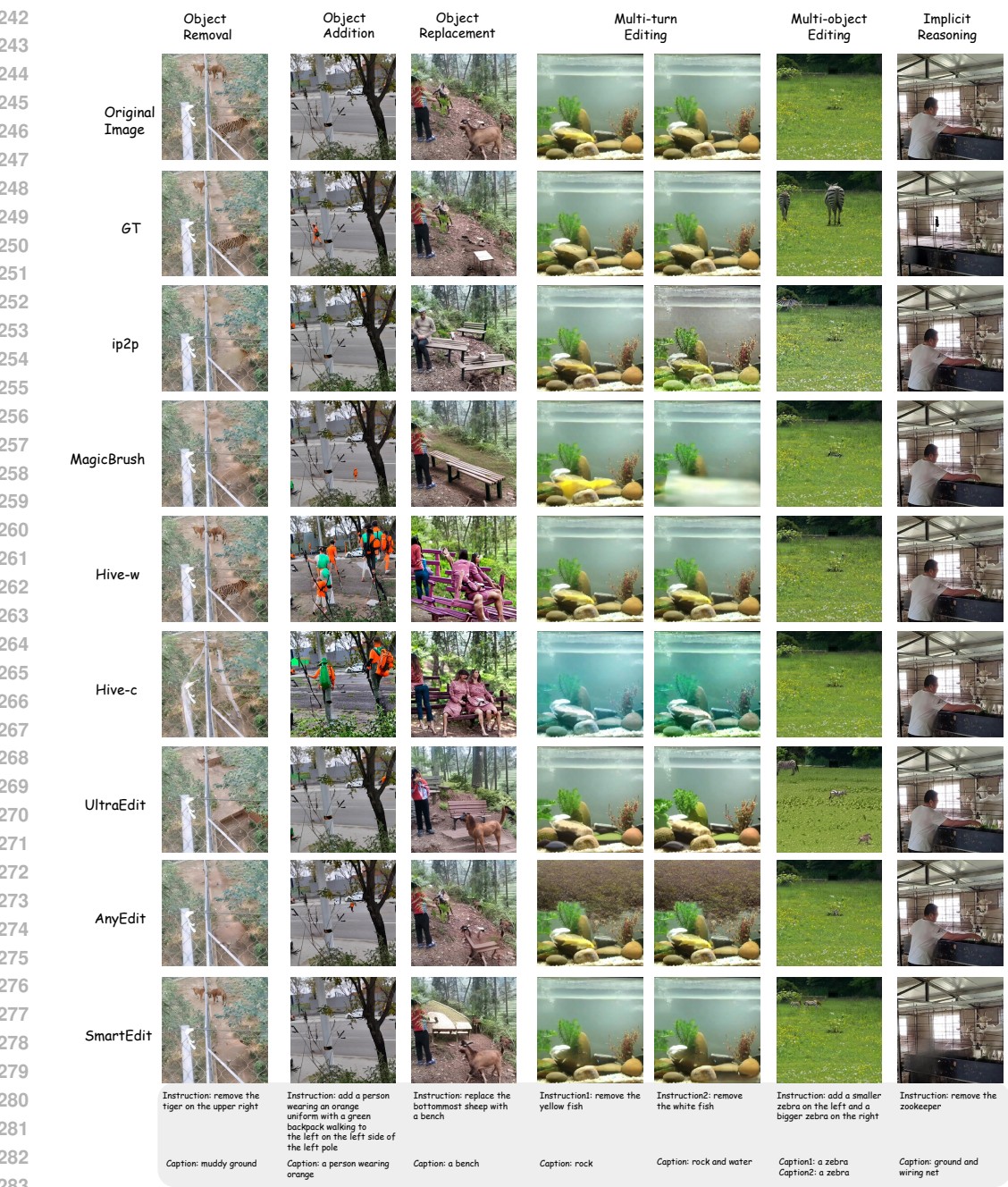

Figure 15: **Cases of Local Editing Results.**

mation. All annotators underwent extensive training, including exposure to exemplary cases, and received iterative feedback to progressively enhance their performance. To ensure annotation fidelity and inter-annotator consistency, the quality assurance specialist conducted systematic evaluations of all outputs. Significantly, all procedures were implemented in a double-blind framework to mitigate potential experimental biases.

**Participant Disclosure and Consent.** Annotators were comprehensively briefed regarding the purpose of the annotation task, their unconditional right to withdraw participation at any juncture, and the exclusive research-oriented utilization of their annotations. No personally identifiable information was collected during this process. All annotators provided documented informed consent prior to their participation in the study.

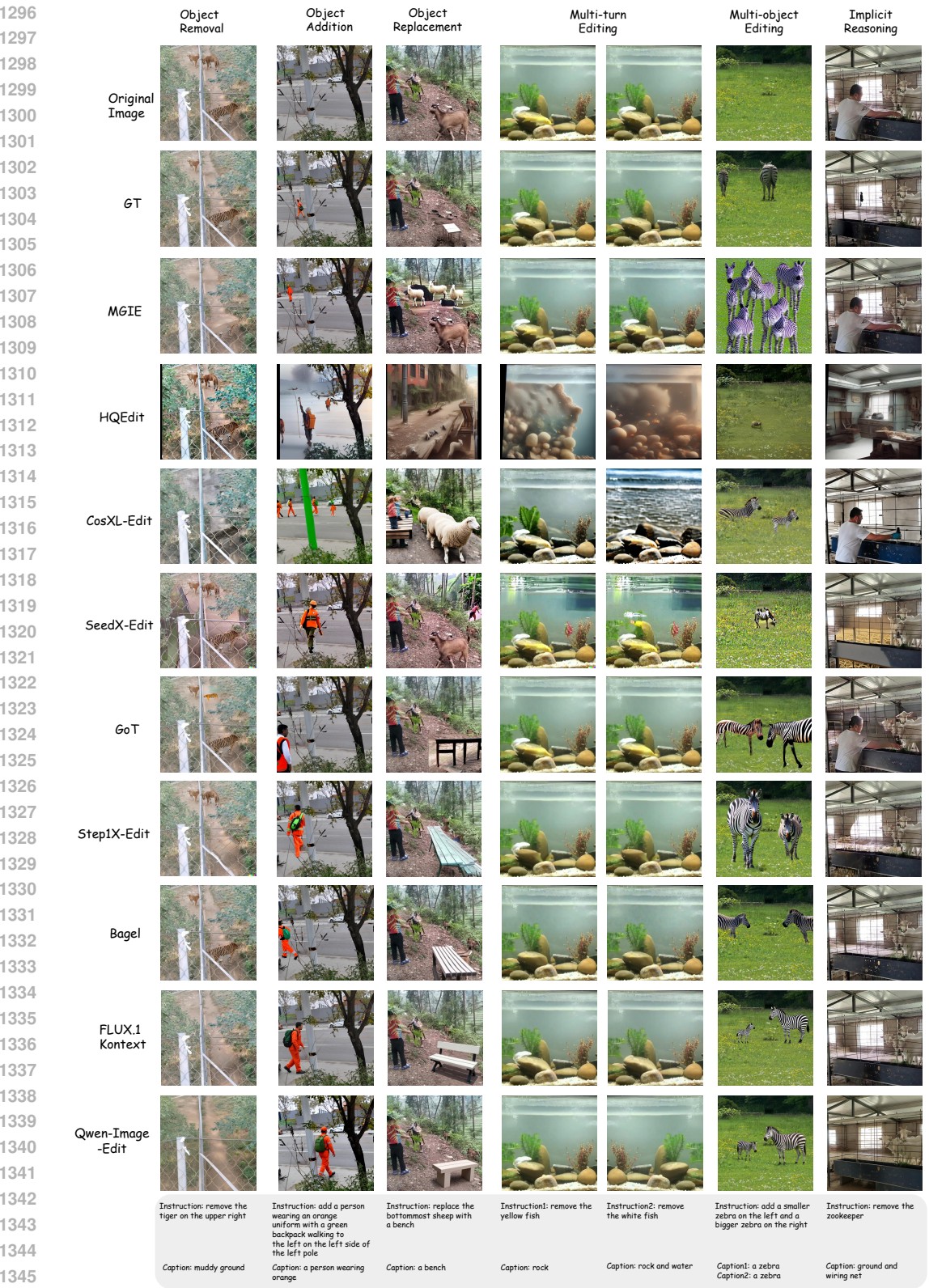

Figure 16: **More Cases of Local Editing Results.**

**Compensation.** Annotators received remuneration at a rate commensurate with or exceeding the local minimum wage standards, in strict adherence to the ICLR Code of Ethics regarding fair compensation practices.

|  | action | location | view |
|---|---|---|---|

**AnyEdit**

instruction: bend down the head of the left parrot
score: 0
reason: The edited image does not show any evidence of the left parrot's head being bent down. Instead, the beaks of all parrots have been altered, which is unrelated to the specified action. The intended action of bending down the head is not implemented at all.

instruction: move the giraffe to the left of the tree
score: 0
reason: The images are identical except for the removal of the giraffe, which does not fulfill the instruction to move the giraffe to the left of the tree. No attempt to relocate the specified object is evident.

instruction: shift the view upward until sky can be seen
score: 0
reason: No proper viewpoint transformation was detected. The two images appear to depict different scenes entirely, as the landscape and vegetation differ significantly between the images.

**Step1X-Edit**

instruction: have a grey cat step towards the yellow cat
score: 9
reason: The grey cat is positioned naturally as if stepping towards the yellow cat, with appropriate contact points on the floor. The lighting and shadows are consistent with the scene, and the interaction with the surroundings is well-maintained. The only minor issue is the lack of motion blur, which could enhance the realism of the movement.

instruction: remove the objects in the sky
score: 9
reason: The objects in the sky have been successfully removed, and the background consistency is well maintained. The sky appears natural, and there are no noticeable artifacts or inconsistencies in lighting or perspective. The integration is nearly seamless, with only minor potential for improvement in blending.

instruction: shift the view to the right a little
score: 8
reason: The viewpoint transformation is executed well, with a slight shift to the right as intended. The spatial relationships between objects remain consistent, and the perspective and foreshortening are preserved. The lighting and shadows are consistent with the new viewpoint, and textures and details are maintained across surfaces. However, the change is subtle, which may not fully demonstrate the intended shift.

**Smart-edit**

instruction: make the white boat fall into water with a splash
score: 8
reason: The edited image successfully depicts the white boat having fallen into the water, as evidenced by the large splash. The splash appears natural and consistent with the action of a boat hitting the water. The scene maintains visual coherence, with the splash integrated well into the water's surface. However, the absence of the white boat in the splash area could be more pronounced to enhance realism. Overall, the transformation is well-executed with minor room for improvement in depicting the boat's interaction with the water.

instruction: make the girl ride her bicycle away
score: 6
reason: The girl has been successfully added to the scene, riding a bicycle. The integration of the girl and the bicycle into the scene is moderately successful. The lighting on the girl and the bicycle is consistent with the scene, and the perspective is reasonably maintained. However, the size of the bicycle wheels is disproportionately large compared to the rest of the scene, which affects the realism. The background consistency is well-maintained, with no noticeable disruptions. Overall, the integration is decent but could be improved with better proportion adjustments.

instruction: shift the view upward until the road no longer be seen
score: 0
reason: The two images do not depict the same scene from different viewpoints. They appear to be entirely different locations, and no proper viewpoint transformation is evident.

**GoT**

instruction: let the elephant take a step to the right
score: 3
reason: The edited image shows some attempt at motion with the introduction of motion blur, suggesting movement. However, the elephant's position relative to the background and its own body parts does not convincingly depict a step to the right. The motion blur is excessive and not well-aligned with the direction of the intended movement, and there is no clear evidence of a shift in the elephant's position or interaction with the ground that would indicate a step. The scene lacks proper shadow and contact point adjustments to support the action.

instruction: shift the position of the man to the left
score: 0
reason: The images do not show evidence of the same man being shifted to the left. Instead, a different person is present in the edited image, indicating no attempt to move the specified object as instructed.

instruction: shift the view to the left until the two riders on the right no longer be seen
score: 2
reason: The edited image attempts to shift the viewpoint to the left, removing the two riders on the right. However, the transition is poorly executed, as the area where the riders were is not properly filled in with consistent textures or details, resulting in an unrealistic appearance. The spatial relationships and perspective are not convincingly maintained, and the lighting and shadows do not adapt correctly to the new viewpoint.

**MGIE**

instruction: let the yellow cat step close to the toy on the floor
score: 0
reason: The edited image does not show any realistic movement of the yellow cat towards the toy. Instead, it features cartoonish yellow cat figures added to the scene, which do not represent the intended action of the real yellow cat stepping closer to the toy. There is no evidence of the described action being implemented in a realistic manner.

instruction: make the man pushing the bicycle closer to another bicycle a little
score: 0
reason: The images do not show any evidence of the man pushing the bicycle being moved closer to another bicycle. Instead, a new person on a bicycle has been added to the scene, which does not fulfill the instruction given.

instruction: shift the view backward
score: 0
reason: No proper viewpoint transformation was detected. The images appear to show different scenes entirely, with the first image depicting a road under an overpass and the second image showing a stylized motorcycle design.

Figure 17: **Action, Location and Viewpoint Editing Examples.**

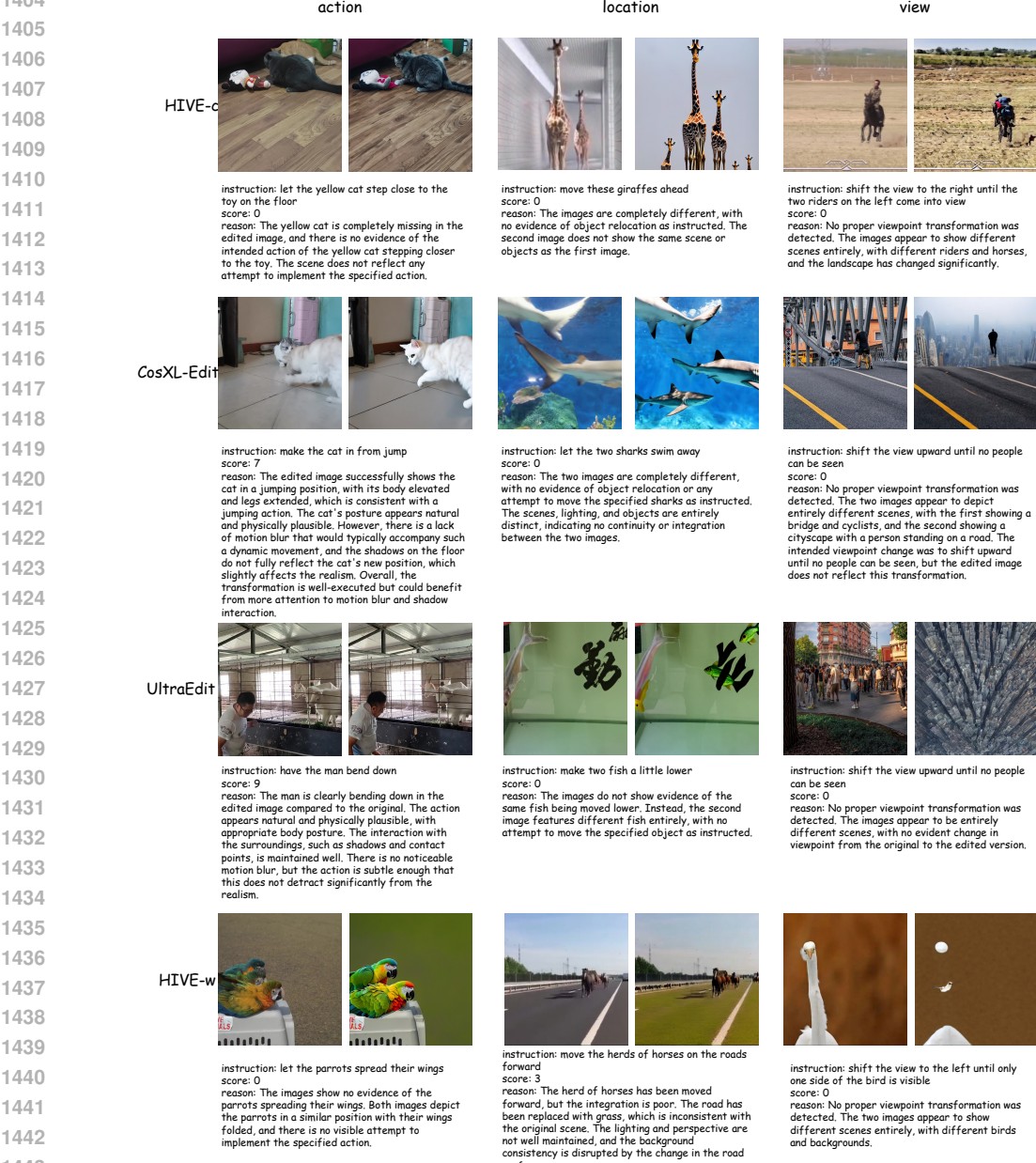

Figure 18: **Action, Location and Viewpoint Editing Examples.**

# I  STATEMENT OF LIMITATION, ETHICAL CONCERN AND BROADER IMPACT

**Limitation.** A significant constraint of our methodology resides in the substantial computational overhead and procedural complexity of the integrated pipeline. The framework encompasses multiple sequential stages of model-based processing followed by comprehensive human evaluation, necessitating considerable computational resources and expert human intervention. This intricate design imposes reproducibility challenges, hampers scalability, and restricts the exhaustive coverage of the complete spectrum of compositional editing tasks. Furthermore, components leveraging large language models are inherently bounded by contemporary model limitations, potentially compromising performance on tasks demanding sophisticated reasoning or precise visual-linguistic integration.

**Ethical Considerations.** While our instruction-based image editing framework demonstrates robust capabilities, it elicits ethical considerations regarding potential misappropriation and privacy implica-

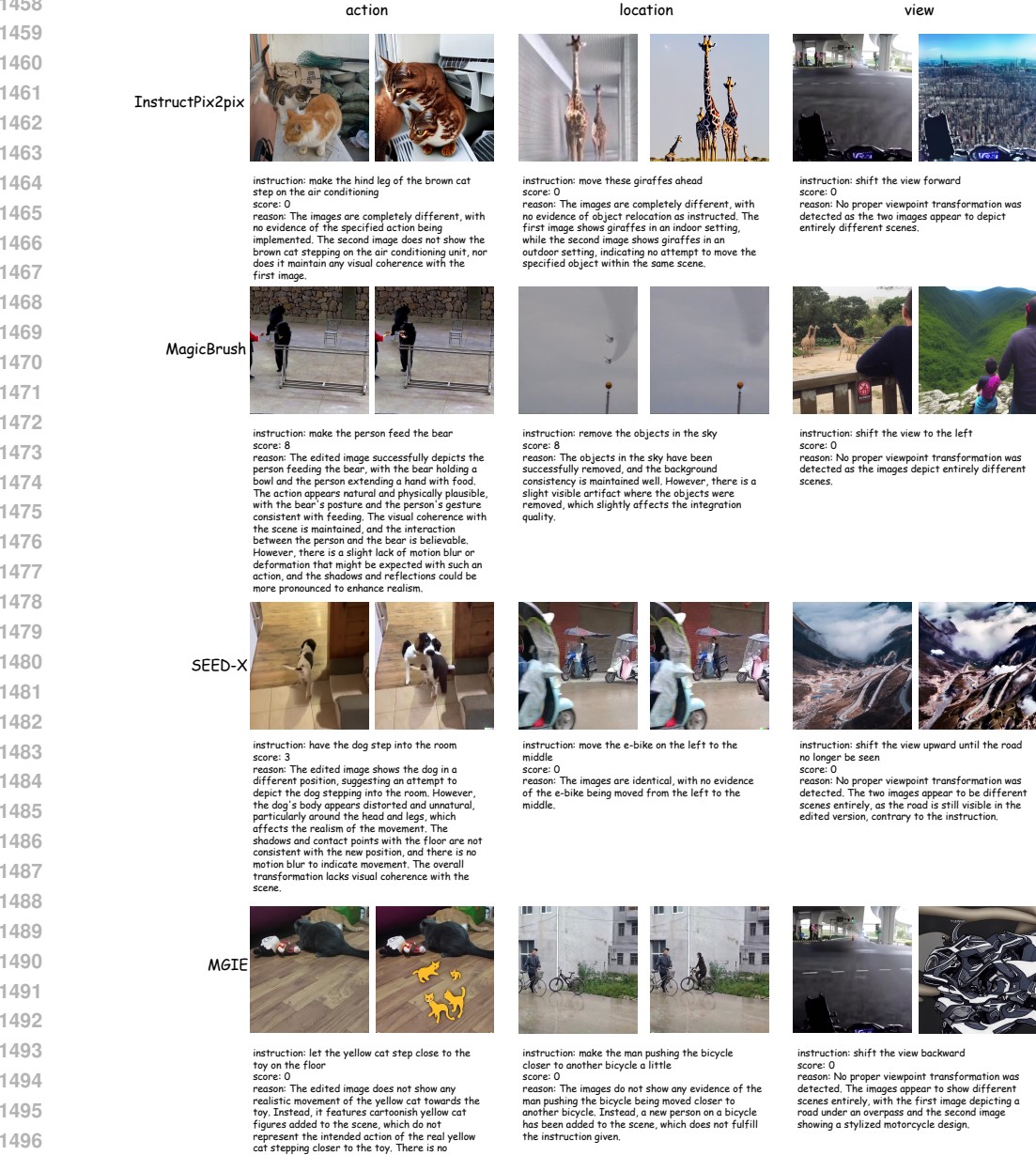

Figure 19: **Action, Location and Viewpoint Editing Examples.**

tions. The capacity to generate photorealistic manipulations may be exploited to create deceptive content, potentially exacerbating misinformation dissemination if inadequately regulated. Moreover, the utilization of real-world imagery introduces non-negligible risks of inadvertently exposing personally identifiable information, despite rigorous anonymization protocols. Ensuring responsible deployment necessitates implementing comprehensive safeguards, including content verification mechanisms, stringent data protection frameworks, and transparent operational guidelines.

In line with these considerations, we affirm our adherence to the ICLR Code of Ethics. This work does not involve research with human subjects; accordingly, institutional review board (IRB) approval was not required. Where crowd-sourced annotation was used, the full participant instructions and compensation details are included in Appendix H. All third-party assets (datasets, code, and models) are properly credited with their licenses, and any new assets introduced in the paper are documented to enable responsible reuse. We assess our artifacts as posing no special high-risk dual-use concerns

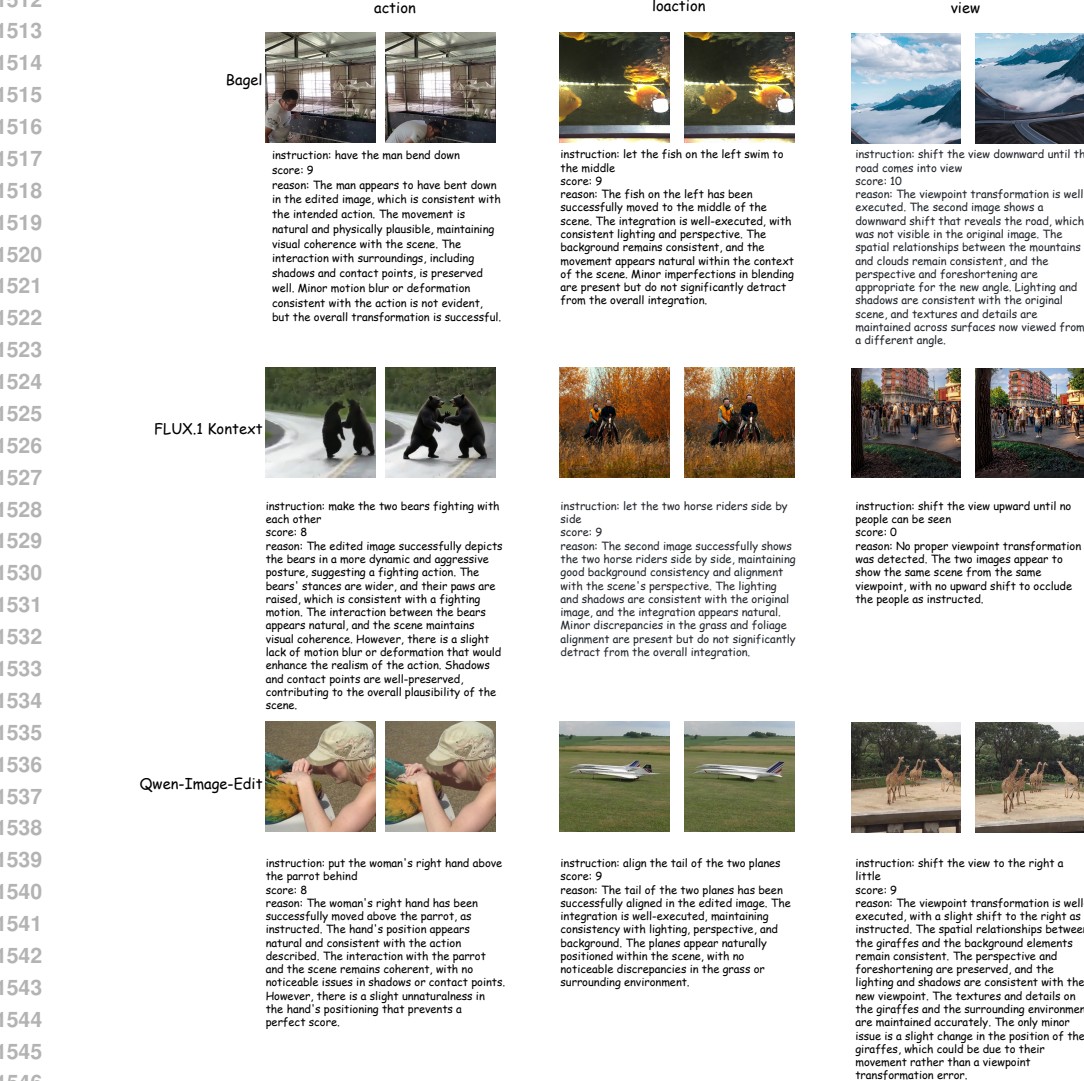

Figure 20: **Action, Location and Viewpoint Editing Examples.**

that would require additional safeguards upon release. We also disclose substantive uses of large language models that materially affect our methods. This discussion of potential positive and negative societal impacts aims to promote responsible interpretation and deployment.

**Broader Impact.** CompBench establishes a rigorous evaluation framework for instruction-guided image editing, facilitating systematic assessment of complex editing capabilities in multimodal large language models. This contribution will accelerate the development of reasoning-aware and controllable editing systems, enhance model performance across visual understanding and generation domains, and expand the practical applicability of large-scale models in diverse real-world contexts, including creative design processes, digital content production, and interactive artificial intelligence assistants.

## J  REPRODUCIBILITY STATEMENT

We take reproducibility seriously and provide clear pointers to the information needed to re-create our results. The benchmark data are open-sourced, and code (with scripts, environment specifications, and instructions for data access/preparation and exact run commands) will be released following submission; these materials are intended to faithfully reproduce the main experiments. The paper discloses the experimental setup, including data splits, hyperparameters, and optimizer choices, in

Section 4 and Appendix A. We also report compute resources (worker types, memory, and execution time) in Appendix A to facilitate environment matching. In addition, we explicitly note where the information needed to reproduce the primary empirical results can be found Section 4; no separate theoretical results are claimed.

## K    THE USE OF LARGE LANGUAGE MODELS (LLMs)

In accordance with the ICLR policy on LLM usage, we disclose that LLMs were used primarily as general-purpose writing assistants for language polishing. Their role was limited to correcting grammar and punctuation, improving clarity and flow. LLMs did not contribute to research ideation or problem formulation; model or algorithm design; dataset creation or labeling; experiment setup, tuning, or analysis; drafting of substantive technical content; or code/results generation. All scientific claims, methods, and conclusions were conceived, written, and verified by the authors. No proprietary or sensitive data were provided to the LLM service. Given this limited, editing-only role, the LLMs should not be regarded as contributors.

