# OpenReview forum: "CompBench: Benchmarking Complex Instruction-guided Image Editing"
_ICLR.cc/2026/Conference — ICLR 2026 Conference Withdrawn Submission_

### Official Review · Reviewer_yTjJ · 2025-10-27

**Soundness:** 3
**Presentation:** 3
**Contribution:** 3
**Rating:** 6
**Confidence:** 3

**Summary:**

This paper introduces CompBench, a comprehensive benchmark for evaluating complex instruction-guided image editing models. The benchmark addresses limitations in existing datasets by providing more realistic scene complexity, comprehensive task coverage, and high-quality data curation. CompBench features 3,000+ image editing pairs across 9 tasks organized into 5 major categories: local editing, multi-editing, action editing, scene spatial editing, and complex reasoning. The authors propose an MLLM-human collaborative framework for data construction and an instruction decomposition strategy to enhance clarity. Extensive evaluation of 16 state-of-the-art models reveals that MLLM-based architectures and reasoning capabilities are crucial for superior performance.

**Strengths:**

+ The paper addresses a  gap in existing benchmarks by focusing on complex, realistic editing scenarios that better reflect real-world applications. The instruction decomposition strategy along four dimensions (location, appearance, dynamics, objects) is well-motivated. The MLLM-human collaborative framework for data construction represents a creative approach to ensuring high-quality annotations.
+ The benchmark construction methodology is rigorous, involving multiple stages of quality control including automated filtering, MLLM evaluation, and human verification. The evaluation framework is comprehensive, employing both automatic metrics (PSNR, SSIM, LPIPS, CLIP scores) and human evaluation with GPT-4o and Qwen-VL for complex tasks. The statistical analysis of scene complexity using quantitative metrics (average objects, categories, occlusion rates) provides solid evidence of increased difficulty.
+ The paper is well-structured with clear explanations of the benchmark construction pipeline, task categorization, and evaluation methodology.

**Weaknesses:**

+ While 3,000+ samples represent a substantial effort, the benchmark is relatively small compared to some existing datasets (e.g., UltraEdit with 4M samples). The reliance on MOSE video dataset as the primary source may introduce domain bias, potentially limiting the diversity of visual scenes and contexts.
+ The heavy reliance on CLIP-based metrics for foreground evaluation may inherit known limitations of CLIP in understanding fine-grained visual details and spatial relationships. The use of GPT-4o for evaluation, while innovative, introduces potential inconsistencies and may not always align with human judgment. The paper would benefit from more extensive validation of these automatic evaluation metrics against human annotations.
+ The allocation of samples across tasks is highly uneven (e.g., 1331 for object removal vs. 73 for location editing). While the authors justify this based on task complexity, it may lead to unreliable performance estimates for tasks with fewer samples and limit the statistical significance of comparisons.
+ The paper lacks detailed analysis of common failure modes and their underlying causes. Understanding why certain models fail on specific tasks would provide more actionable insights for model development.

**Questions:**

+ How well do the automatic evaluation metrics (especially CLIP-based ones) correlate with human judgment across different task types?
+ What was the inter-annotator agreement during the human verification process? How did you handle cases where annotators disagreed on instruction quality or editing success?

---

### Official Review · Reviewer_XzSv · 2025-10-27

**Soundness:** 2
**Presentation:** 2
**Contribution:** 1
**Rating:** 2
**Confidence:** 4

**Summary:**

The paper introduces CompBench, a challenging benchmark for instruction-guided image editing with around 3K samples across multiple categories, but it is not truly large-scale or highly diverse compared to existing M-level datasets. While it highlights limitations of current models on difficult editing tasks, it does not propose preliminary solutions or demonstrate how using this dataset could lead to meaningful improvements, which is a major concern.

**Strengths:**

The paper presents a dataset with a rich variety of categories and provides preliminary experiments demonstrating this diversity, confirming certain limitations of existing instruction-guided image editing models.

**Weaknesses:**

1. Limited dataset scale and utility: The dataset contains only ~3K samples, far from the claimed large-scale, and is better characterized as a test set rather than a full dataset. Its size and diversity are insufficient to support training new models or demonstrate substantial improvements.

2. Lack of methodological contribution: The work does not propose any new models or preliminary solutions leveraging the dataset. Merely highlighting the limitations of existing models without offering methods to address them is a critical shortcoming.

3. Limited novelty and impact: By only evaluating existing models on a small test set, the work provides limited scientific or technical insight and does not follow the typical dataset paper paradigm, which normally combines a new dataset with methods that demonstrate its utility.

**Questions:**

1. Can a dataset of only 3K samples really be considered large-scale? Given the task difficulty and the diversity of categories, the dataset does not seem sufficiently large to effectively train a high-quality image editing model.

2. For the newly proposed dataset, it is unclear whether any effective models exist—does the paper propose a method capable of addressing the challenges posed by this dataset?

3. Based on your new dataset, it is unclear what improvements existing models can achieve and whether the dataset can meaningfully enhance their performance.

4. Is there any experiment showing that, with support from your new dataset, existing models can achieve better performance on previous tasks?

5. Current instruction-guided image editing models clearly have limitations, and the difficulty of the task is already well-known—there is no need for a new dataset to demonstrate this. Merely highlighting an issue that is widely recognized, without proposing any corresponding solutions, raises questions about the significance of the contribution.

---

### Official Review · Reviewer_HxsN · 2025-11-02

**Soundness:** 2
**Presentation:** 3
**Contribution:** 2
**Rating:** 4
**Confidence:** 5

**Summary:**

This paper presents CompBench, a large-scale benchmark specifically designed to evaluate complex instruction-guided image editing systems. CompBench is constructed through a multi-stage MLLM-human collaborative pipeline, carefully curated to encompass tasks reflecting real-world complexity: spatial reasoning, appearance manipulation, multi-object operations, action/dynamics edits, and implicit/complex reasoning. The benchmark features a diverse taxonomy of nine task types across five categories, includes detailed annotation protocols, and emphasizes fine-grained, multi-dimensional instructions.

**Strengths:**

1. CompBench covers nine diverse editing tasks, ranging from local and multi-object edits to action, viewpoint, and implicit reasoning tasks.
2. The data pipeline leverages both MLLMs and repeated expert human verification.

**Weaknesses:**

1. The positioning with respect to the very latest complexity-aware or dynamic-editing benchmarks is incomplete. For example, ByteMorph [1] and ComplexBench-Edit [2], both of which focus on complexity-controllable/dynamic edits.

2. The work claims to cover 'action editing' and dynamic scene manipulations, yet the quantitative detail and qualitative discussion are not as deep or exhaustive as for local/multi-object cases. The action/location/viewpoint results are summarized in only a single table, with little ablation or error analysis.

3. Several highly relevant and recent benchmarks and evaluation protocols are not cited or discussed: ByteMorph [1], ComplexBench-Edit [2], EditInspector [3], RefEdit [4], KRIS-Bench [5], MedEBench [6], and ADIEE [7]. These papers directly pertain to comprehensive evaluation of instruction-guided editing and introduce new metrics, protocols, or specialized domains (e.g., referring expressions, knowledge-based reasoning, medical imaging). Their omission weakens the positioning, and some of these works should be included both in the Related Works and in comparative/limitations discussion (e.g., applicability and task overlap).

4. The paper outlines a structured "instruction decoupling"/dimension-aware pipeline, but offers limited empirical evidence for the impact of this method versus prior annotation protocols.

5. Several implementation nuances (e.g., CLIP version, exact object region selection strategy, normalization scheme for composite scores, handling of multiple edits within an image) are ambiguous in the main text.

6. This is expected for a benchmark paper, but the submission centers almost entirely on benchmark construction and experimental validation, with no core modeling or algorithmic contributions.

7. For some tasks, there are existing strong editing methods (domain-specific or action-based) that are not included as baselines or discussed in the text, such as methods for explicit "referring expression edits" or specialized medical-editing pipelines (e.g., RefEdit [4], KRIS-Bench [5], MedEBench [6]).


[1] ByteMorph: Benchmarking Instruction-Guided Image Editing with Non-Rigid Motions

[2] ComplexBench-Edit: Benchmarking Complex Instruction-Driven Image Editing via Compositional Dependencies

[3] EditInspector: A Benchmark for Evaluation of Text-Guided Image Edits

[4] RefEdit: A Benchmark and Method for Improving Instruction-based Image Editing Model on Referring Expressions

[5] KRIS-Bench: Benchmarking Next-Level Intelligent Image Editing Models

[6] MedEBench: Diagnosing Reliability in Text-Guided Medical Image Editing

[7] ADIEE: Automatic Dataset Creation and Scorer for Instruction-Guided Image Editing Evaluation

**Questions:**

see Weaknesses

---

### Note · Authors · 2025-11-13

I have read and agree with the venue's withdrawal policy on behalf of myself and my co-authors.